# Integrated linkage-driven dexterous anthropomorphic robotic hand

Uikyum Kim [1,2,5 ✉], Dawoon Jung[3,5], Heeyoen Jeong[4], Jongwoo Park [2], Hyun-Mok Jung[2], Joono Cheong [3], Hyouk Ryeol Choi[4], Hyunmin Do [2] & Chanhun Park [2]

Robotic hands perform several amazing functions similar to the human hands, thereby offering high flexibility in terms of the tasks performed. However, developing integrated hands without additional actuation parts while maintaining important functions such as human-level dexterity and grasping force is challenging. The actuation parts make it difficult to integrate these hands into existing robotic arms, thus limiting their applicability. Based on a linkage-driven mechanism, an integrated linkage-driven dexterous anthropomorphic robotic hand called ILDA hand, which integrates all the components required for actuation and sensing and possesses high dexterity, is developed. It has the following features: 15-degree-of-freedom (20 joints), a fingertip force of 34N, compact size (maximum length: 218 mm) without additional parts, low weight of 1.1 kg, and tactile sensing capabilities. Actual manipulation tasks involving tools used in everyday life are performed with the hand mounted on a commercial robot arm.

[1] Department of Mechanical Engineering, Ajou University, Suwon 16499, Korea. [2] Department of Robotics and Mechatronics, Korea Institute of Machinery & Materials (KIMM), Daejeon 34103, Korea. [3] Department of Control and Instrumentation Engineering, Korea University, Sejong 30019, Korea. [4] Department of Mechanical Engineering, Sungkyunkwan University, Suwon 16419, Korea. [5]These authors contributed equally: Uikyum Kim, Dawoon Jung. ✉email: ukim@ajou.ac.kr

Interpreting the extremely complex functioning of the human hand remains an unresolved challenge in the field of robotics[1,2]. In particular, the movement of the human hand involves considerably high dexterity levels, suitable for performing a wide variety of tasks requiring a strong gripping force ranging from fine object grasping to tool manipulation[3,4]. Out of the 206 bones in the human body, 54 bones are in the hands, corresponding to a quarter of the total number of bones; the muscle structure driving them is also extremely complex. In addition, the tactile corpuscles, which enable tactile sensation, are mostly distributed in the hand, and they help in performing delicate tasks[5]. In particular, because most tactile corpuscles are distributed at ~1 mm intervals in the fingertips, delicate tasks are easily performed with the fingertips[6].

To implement these functions using robots, many dexterous anthropomorphic robotic hands have been developed. For performing efficient grasping motions, many effective robotic hands in a form capable of adaptive grasping or low degree of freedom (DOF) have been developed[7–12]; however, our analysis focused on multi-DOF hands with high dexterity. Therefore, the representative core mechanisms of the dexterous robotic hand are classified into (1) motor-direct-driven, (2) tendon-driven, and (3) linkage-driven mechanisms.

The hands developed based on the motor-direct-driven mechanism are structures that intuitively position the motors with respect to the joints to drive the joints directly or using a gear or timing pulley[13,14]. Such a structure may have high joint driving efficiency, and it is easy to arrange the joints at a desired position. In particular, MPL v2.0, which was developed by Johns Hopkins APL, shows a high dexterity with active 22 DOF and a compact design integrating actuators and electronics. This hand is capable of human-level natural movement and tactile feedback[15]. However, the size and performance of the hands are highly dependent on the motor, especially the finger part. Using motors with high-end specifications or driving force transmission parts result in increased costs. In addition, the inertia at the finger is high owing to the weight of the motor, thus requiring complex control mechanisms. Further, the space between the fingers is narrow, which makes it difficult to wire the force sensor to the finger. Therefore, it is difficult to achieve compactness and high performance without innovation in actuator technology.

The hands based on the tendon-driven mechanism are the most similar to the human driving mechanism. In general, their actuators are located on the forearm and connected to the joints by tendons to transmit the driving force[16–21]. The Robonaut hand[16] developed by NASA, David hand[17] developed by DLR, and Shadow dexterous hand[18] developed by Shadow Robot Company can be considered as representative hands with such a mechanism. It is possible to realize movements almost similar to those of the human hand and generate a high fingertip force depending on the connection configuration of the tendon. It is a highly suitable approach for developing a single humanoid robot. However, it is very difficult to combine these robotic hands with many existing commercial robot arms or those under development[22–27] because the actuators for driving the robot and the electric parts are attached in the form of a fairly large forearm. For independent driving of joints where several tendons are connected, the tendons require a structure that traverses the axes of rotations of the joints or a special tendon connection structure. This creates assembly and maintenance complexities, which lead to an increase in cost, and even if one tendon is broken or released, it is difficult to repair it. In addition, if two tendons are connected to one joint for antagonistic mechanisms like human joint driving, pre-strain should be applied to the tendon, which leads to a decrease in driving efficiency due to increased friction. Further, when only one tendon is connected to one joint, a return spring is installed to move in the opposite direction. In this mechanism, force control on the spring side becomes difficult.

The linkage-driven mechanism is commonly used in our daily lives. The hands developed based on this mechanism facilitate joint movement in the desired direction through a structure that combines several links transmitting power from the actuator[28–31]. The Schunk SVH 5-finger hand[28] developed by Schunk can be considered as a representative example for this mechanism. It includes a combination of a simple bar and cylindrical links, and has advantages such as bidirectional control of joints, robustness, and ease of manufacture and maintenance[32]. However, it is difficult to implement multi-DOF motions and maintain a large workspace with this mechanism, especially in a serial manipulator such as that used in a finger. The tendon is thin and flexible, so it is possible to drive each joint independently through the axis of rotation, but the link is relatively thick and hard, making this configuration difficult to implement. Therefore, this configuration is frequently used in parallel mechanisms such as Stewart platforms or delta robots and in mechanisms with low DOF[33].

Based on the above analysis, we concluded that it is important to have the following advantageous features in a robotic hand: dexterity, fingertip forces, controllability, robustness, low cost, low maintenance, and compactness. The definitions of these features are provided in Supplementary Text 1[34–37]. In addition, it is necessary to develop a robotic hand in which all parts are embedded in the hand itself and includes all the above functions.

In the present study, an integrated linkage-driven dexterous anthropomorphic robotic (ILDA) hand was developed. To include the abovementioned features, a finger mechanism was developed for the robotic hand. The mechanism was constructed through the fusion of parallel and series mechanisms to implement 2-DOF movements in a metacarpophalangeal (MCP) joint and 1-DOF movements in a proximal interphalangeal (PIP) joint through link combinations. The design was developed considering the selection of small parts that can play the role of each joint, part placement and configuration to achieve the desired DOF motion and drive angles, and an efficient power transmission structure to obtain a high fingertip force and its back drivability. In addition, the force-sensing capability of the hand is secured by attaching a six-axis force/torque (F/T) sensor to the fingertip. With the designed finger, a robotic hand having 15 DOF and 20 joints with five fingers were developed. For practical use, it was constructed by resolving the board layout and wiring problems to ensure the compactness of the electronics. All motors were integrated in the palm of the hand having five fingers with fingertip sensors (Fig. 1a, b). Therefore, it could be easily attached to a general robot arm with a simple connection configuration, as shown in Fig. 1c. To evaluate the performance of the developed hand, its performance was analyzed through several experiments. Experiments were conducted to determine the possibility of grasping objects with various shapes, provide strong grasping forces to crush cans, and ensure delicateness when holding eggs. Finally, the high utilization possibility of the hand was verified through tests involving cutting paper with scissors and picking up small objects with tweezers, thus replicating tool operations performed in everyday life (Fig. 1d).

## Results

**Design of linkage-driven mechanism.** Linkage-driven mechanisms have inherent advantages such as ruggedness and ease of manufacture and maintenance; however, they also have disadvantages such as difficulty in performing multi-DOF movements in a small space while maintaining sufficient workspace. Therefore, it is important to implement a linkage-driven robotic finger mechanism with human-finger-like 3-DOF movements

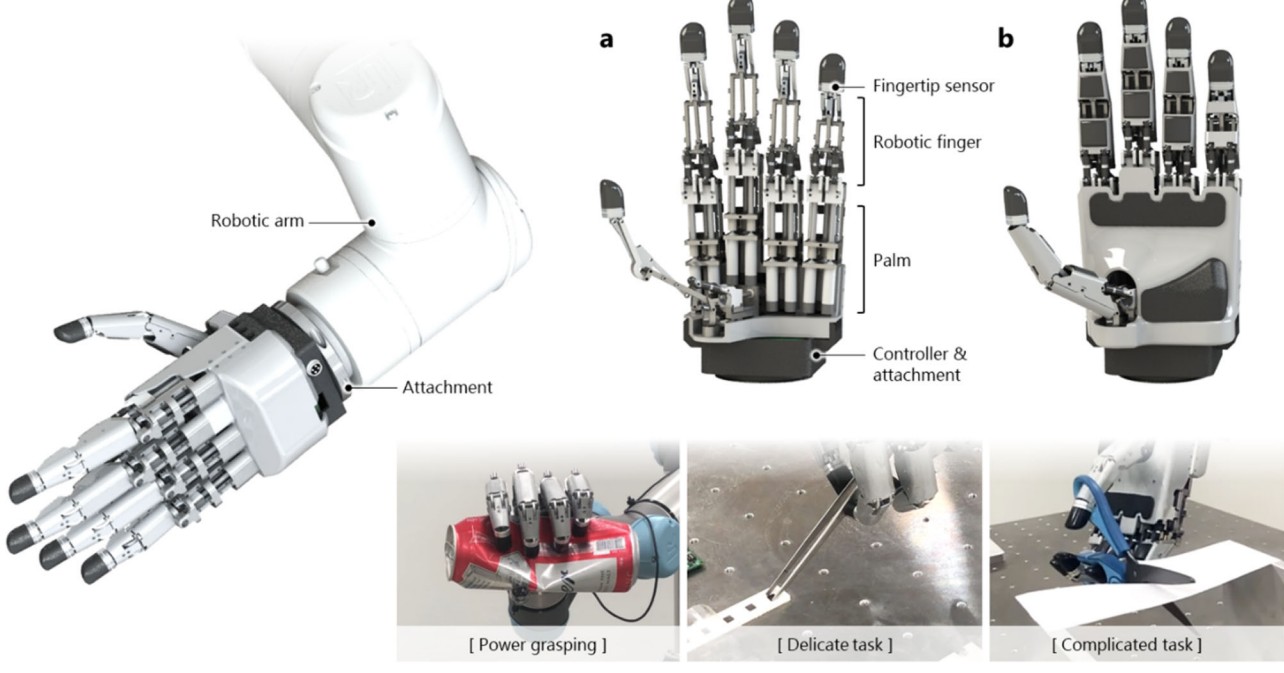

**Fig. 1 Overview of the ILDA. a** Configuration of the ILDA hand composed of five robotic fingers with fingertip sensors, the palm side integrating the actuators, and the controller and attachment. **b** ILDA hand with covers. **c** Ease of attachment of the ILDA hand to a developed robotic arm. **d** Actions performed using the ILDA hand such as grasping and manipulating everyday life tools, showing delicate and strong grasping.

having a narrow finger-sized workspace to ensure the dexterity of the robotic hand. To overcome the aforementioned shortcomings and include advantageous features, the mechanism of the robotic finger was designed according to the schematics shown in Fig. 2.

The kinematic design for the finger of a robotic hand (Fig. 2a) is shown in Fig. 2b. For structural simplicity, three serial chains were placed on the same ground surface. The two PSS (prismatic-spherical-spherical) chains on the front constitute a 2-DOF parallel mechanism with one universal joint, which generates a virtual triangular structure. A vertex of the triangle was fixed to the universal joint that performs a 2-DOF MCP motion. To implement a single DOF of the PIP joint, the PIP joint must be driven independently of the MCP joint. Most linkage-driven robotic fingers have realized only 1- or 2-DOF movements in which the two joints are subordinated. As a result, a linkage-driven robotic finger with 3-DOF has not been investigated so far, as shown in Supplementary Table 1[38–43].

The prismatic-spherical-universal (PSU) chain passes through the inside of the triangle. The V-shaped bell-crank part at the end of the chain is connected to one side of the triangle as a revolute joint. Here, the bell-crank part enables efficient transmission of the driving force. Detailed information about the bell-crank design is provided in Supplementary Text 3[44–46]. A crossed four-bar linkage is connected in series to the bell-crank part to enable a 1-DOF PIP motion. Constructing an additional four-bar linkage couple the movement of the PIP and DIP joints. Therefore, the entire mechanism is composed of a 2-DOF MCP motion generated by the parallel mechanism and a single-DOF PIP generated by the serial mechanism. Through linear displacements ($d_1$, $d_2$, and $d_3$) at the three prismatic joints, combinations of the 3-DOF motions of the finger can be formed, as shown in Fig. 2c. To help understand the motion of the finger for each DOF, the independent motion of each joint was analyzed (Fig. 2d–f). During one linear motion ($d_3$), the PSU chain moved down, and the bell crank fixed to one side of the triangle rotated to move the

four-bar linkage, thus enabling independent flexion and extension of the PIP joint (Fig. 2d). For flexion and extension of the MCP joint, two linear motions ($d_1$, $d_2$) were created through identical movements in the same direction at the same time (Fig. 2e). At this time, the movements of the MCP do not affect the movement of the PIP joint connected to $d_3$. When $d_1$ and $d_2$ were in opposite directions, the finger performed abduction and adduction movements of the MCP joint (Fig. 2f).

Eventually, the three linear displacements were generated by the combination of the rotating motors and ball screws, and the three motors could simultaneously create the 3-DOF motion and produce a high force output.

**Kinematic analysis of linkage-driven mechanism**. To theoretically analyze the proposed mechanism, we divided the mechanism into four kinematic models (Fig. 3), namely two PSS chains of the MCP joint (Fig. 3a), one PSU chain of the PIP joint (Fig. 3b), one four-bar linkage of the PIP joint (Fig. 3c), and one four-bar linkage of the DIP joint (Fig. 3d). Based on these models, the inverse kinematics of this mechanism was analyzed, which is important for validating the mechanism and controlling its motions.

The parameters used for the inverse kinematic analysis at the MCP joint are shown in Fig. 3a. The global coordinate frame, $O-xyz$, is attached to the ground of the fixed frame. The local coordinate frame, $P-uvw$, is attached to the moving frame on the triangle. At the MCP joint, there are two rotations ($d_1$, $d_2$) related to the flexion/extension and abduction/adduction motions of the finger. Here, the rotation matrix of the moving frame can be written as

$$\mathbf{R} = \begin{bmatrix} Cq_2 & Sq_2Sq_1 & Sq_2Cq_1 \\ 0 & Cq_1 & -Sq_1 \\ -Sq_2 & Cq_2Sq_1 & Cq_2Cq_1 \end{bmatrix}, \quad (1)$$

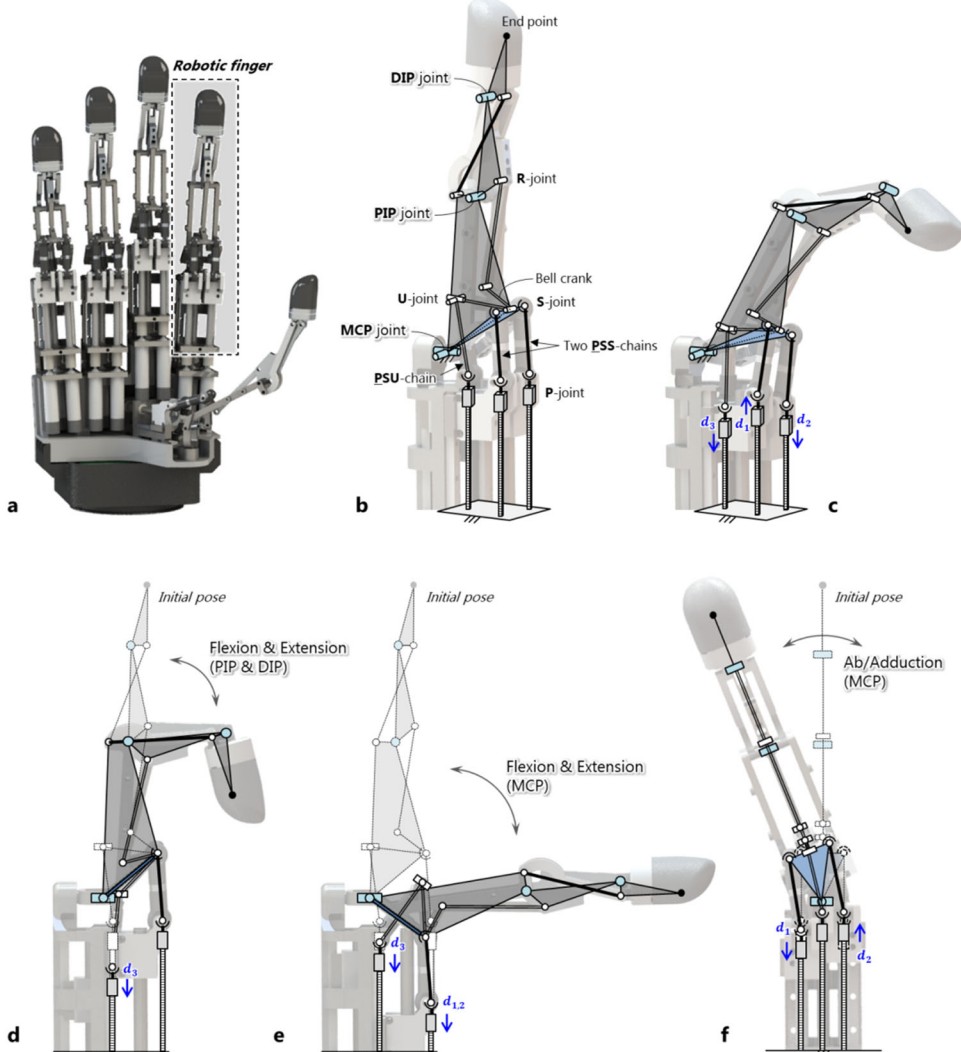

**Fig. 2 Kinematic structure of the proposed mechanism of the robotic finger. a** Mechanism explained in the robotic hand. **b** Kinematic structure of the finger. **c** 3-DOF motion of the finger generated by three linear motions. **d** Flexion and extension of the PIP joint that can be driven regardless of the movement of the MCP joint. **e** Flexion and extension of the MCP joint. **f** Abduction and adduction of the MCP joint.

where $Sq_i$ and $Cq_i$ are the sine and cosine functions of $q_i$, respectively. In Fig. 3a, $A_i$ represents the intersection between the center of the $i$th prismatic joint and the $xy$ plane, and $B_i$ represents the center of the $i$th spherical joint on the moving frame. $d_i$ is the displacement input of the $i$th prismatic joint. From Eq. S2, $d_1$ and $d_2$ are derived as

$$d_i = k_{i,z} \pm \sqrt{k_{i,z}^2 + l_i^2 - \|\mathbf{k}\|_i^2} \quad \text{(for } i = 1, 2), \quad (2)$$

where $d_i$ is calculated using the angular displacement ($\theta_{im}$) of a motor and ball screw. $\theta_{im} = 2\pi d_i/p$, where $p$ is the pitch of the ball screw.

The mechanism related to the motions of the PIP and DIP joints consists of one PSU chain and two crossed four-bar linkages (Fig. 3). Eq. S6 can be summarized in the descending order for $d_3$. Thus, the result is as follows:

$$d_3 = A_1 \pm \sqrt{A_1^2 - B_1}, \quad (3)$$

Because of the additional four-bar linkage, the DIP and PIP joints showed subordinate motions, and their relationships are explained in Supplementary Text S2. Consequently, the inverse

kinematic relations between the angles of the three joints and the three linear displacements can be expressed as a closed-form equation.

To check the workspace of the designed robotic finger, the reachable workspace of the robotic finger was analyzed as shown in Fig. 3e–g. Figure 3e shows the 3D view of the reachable workspace composed of points that can be reached by the end point of the finger. Here, $d_r$ is the distance between the end point of the finger and the fixed MCP joint of the finger, which is taken as the origin. The farthest points are expressed in red. In this case, both PIP and DIP of the finger at this time are stretched poses. As it is easy to adjust the length of the three phalanges of the robot finger, from a design perspective, the length of the phalanges of the human finger was analyzed under the same conditions as those of the robotic finger, for a close comparison. Figure 3f shows the side view of the workspaces of the robotic finger and human finger. Here, $T_1 - T_5$, $T_a$, and $T_{ab}$ indicate the trajectories of the fingers. The trajectory from $T_1$ to $T_2$ shows that the MCP joint moves from full extension (0°) to full flexion (90°) with the stretched pose of the PIP and DIP. The trajectory from $T_2$ to $T_4$ is generated when the PIP and DIP are moved while fixing the MCP joint in full flexion. $T_3$ represents the lowest point on the $z$-axis in

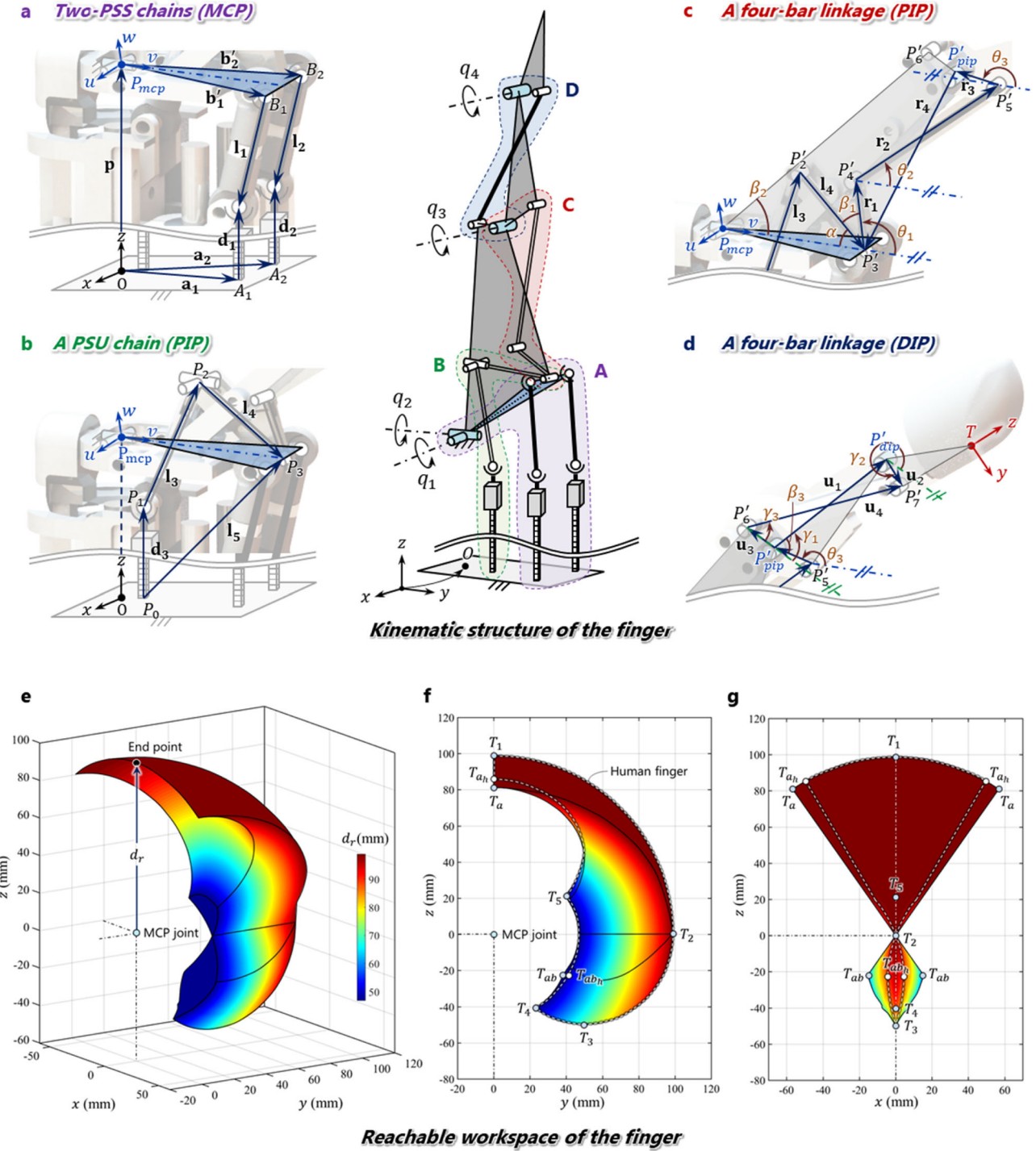

**Fig. 3 Kinematic analysis of the proposed mechanism of the finger. a** Two PSS chains for the 2-DOF motion of the MCP joint. **b** PSU chain for the 1-DOF motion of the PIP joint. **c** Crossed four-bar linkage combined with the PSU chain to drive the PIP joint. **d** Another crossed four-bar linkage to implement subordinate motion of the DIP joint by connecting it with the PIP joint. **e–g** Reachable workspaces of the finger and human finger.

this trajectory. $T_5$ is reached when the MCP joint is fully extended while maintaining full flexion of the PIP and DIP joints. The trajectory of a human finger is indicated by the dotted line. Figure 3f shows a workspace similar to that of the two fingers. The abduction/adduction motion is maximum (30°) when the MCP joint is in full extension and minimum (0°) when the MCP joint is in full flexion[34,47]. For this reason, the workspace volume under the MCP joint is relatively small as shown in Fig. 3g, and a part of its lower right corner becomes sharp as shown in Fig. 3e, g.

The shape of the workspace at the bottom of the robotic finger is similar to that of the human finger. The reason is that when the MCP joint is in full flexion, the abduction/adduction movement is limited because the proximal phalanges contact with the top of the two ball screws as shown in Fig. 2e. $T_a$ and $T_{ab}$ are the locations of the maximum abduction/adduction movement during full extension of each joint of the human and robot fingers, respectively. The robotic hand is capable of 35° movement to perform abduction/adduction. As a result, the workspace

shapes of the robotic finger and human finger are very similar, and the volume of each workspace was found to be 188,740 and 119,800 mm$^3$, respectively. Detailed comparisons of the robotic finger and human finger are provided in Supplementary Table 2[48,49].

**Structure of ILDA hand**. To realize the proposed linkage-driven mechanism, we mainly considered the following factors: (1) Selection and configuration of parts of appropriate size to achieve the desired DOF motion: To realize the function of the above-mentioned kinematic model in the narrow space of a finger shape, it should be properly arranged in the configuration of the model. Therefore, it is important to select small components of a suitable size from a design perspective. (2) Efficient power transmission structure to minimize the friction between the assembly parts. To achieve high fingertip force, a compact structure is required while minimizing the friction force in the power transmission part. (3) Ease of manufacture and assembly. To increase the market penetration of the developed robotic hand, it is also important to evaluate it in terms of cost and maintenance. Therefore, it is important to design a simple and robust robotic hand structure.

Figure 4a presents the designed structure of an anthropomorphic robotic finger. The three joints, fingertips, and palm of the robotic hand are similar to those of the human hand. The hand consists of a fingertip part and a fingertip sensor. On the palm side, there are three motors, three couplings, three ball screws to create linear motions, three LM guides, and a finger frame. The proximal phalanx is moved by the angular displacement of the MCP joint, and the middle phalanx and fingertip are moved by the angular displacement of the PIP joint. Figure 4b shows the exploded view of the palm side of the finger. It is important to obtain high driving efficiency while minimizing the friction between the assembly parts.

An exploded view of the upper portion of the power transmission part is shown in Fig. 4c. The rod end serves as an important joint in the implementation of this mechanism. First, it is possible to continuously rotate it in the axial direction without using the cover of the rod end. The rod end performs the function of a ball joint without limiting the movement of the MCP joint during the full range (0–90°) flexion and extension of the joint. In general, ball bearings allow a limited range due to the contact between its socket and ball. In addition, during the full range of the abduction and adduction (±35°) of the MCP joint, the rod end can act as the ball joint without impeding the movement. Therefore, the 3-DOF motion of the fingers is acceptable and does not interfere with achieving the desired driving angle. Second, if the force output of the instrument is high, it is very important to use parts with high strength that can sustain the force without becoming damaged. Because a link-driven mechanism can be configured as a simple metal shaft or bar, it is easy to develop a highly rigid and sturdy design. Meanwhile, multi-DOF joints can easily have low strength. To withstand high loads on fitness equipment, a rod end is used as the joint in the equipment. Owing to the nature of the ball bearing, it enables movement while withstanding high loads by distributing the load. Therefore, using the rod end makes it possible to achieve high strength.

To realize a robotic hand having five fingers and fingertip sensors with 15 motors, all fingers were designed with the same structure (Fig. 4d). Only the lengths of the thumb and little fingers were different. This configuration is meaningful in terms of the modularity of the fingers, which could enable cost reduction and high utilization potential.

For practical uses of the robotic hand, wiring and board configurations are extremely important. Despite developing an effective robotic hand, its use can be inconvenient if the electronic components are not properly configured. Moreover, in robotic hands, compact wiring of several motors and sensors is necessary. As shown in Fig. 4d, three connection boards, a motor driver board, and a main microcontroller unit (MCU) board constitute the electronics of the robotic hand.

The front and rear views of the developed ILDA hand are shown in Fig. 5a, b. All the power transmission parts and motors are integrated to the palm side of the hand. The five F/T sensors were mounted on each fingertip of the configured finger parts, and the sensor wiring was completed so that it did not interfere with the movement of the finger. Finally, an integrated robotic hand with a maximum length of 218 mm and weight of 1.1 kg was developed. Table 1 summarizes the specifications of the developed hand. Detailed information on the hand dimensions is provided in Supplementary Fig. 3. Information on the actual parts and simplicity of the assembly process are explained in the Methods section.

**Performance evaluation**. To verify the performance of the ILDA hand, we evaluated the (1) dexterity within the workspace, (2) the fingertip force, and (3) tactile sensing capabilities.

Figure 5c–e shows the motions of the five fingers on the hand. The natural motions of the hand are shown in Supplementary Video 1. The working ranges of the finger are shown in Fig. 5f–i. The MCP joint could be driven from 0 to 90° and that the PIP joint could also be operated from 0 to 90°; furthermore, the PIP joint could operate independently of the MCP joint. The abduction and adduction of the finger were confirmed at ±35°. The motion of the finger is shown in Supplementary Video 1. Furthermore, its backdrivability was achieved through efficient driving part design. The torque for the backdrivable motion in the MCP joint was 25.9 mNm, and that for the motion in the PIP joint was 6.3 mNm.

To evaluate the fingertip force and tactile sensing capabilities, the ILDA hand was fixed to the wall, and the finger applied an external force to the hemispherical part located above the commercial reference sensor (Nano25, ATI Industrial Automation) (Fig. 6a). The magnitude of the contact force at the contact point was determined through the fingertip sensor, and the same force was applied to the fingertip and reference sensors. The performance of the fingertip sensor is described in a previous study[50]. The force applied by the fingers was sequentially increased in steps whereas the 25 mA current was increased every 2 s. The magnitude of each force measured by the fingertip and reference sensors was calculated as $F_{mag}$ (= $\sqrt{F_x{}^2 + F_y{}^2 + F_z{}^2}$) to compare the two values (Fig. 6b). The maximum force exerted by the finger was found to be 28 N in the stretched pose and 34 N in the bent pose. The accuracy of the static forces applied by the fingers was verified with an average error of 0.9 N. The values of each dynamic force applied sinusoidally with an amplitude of 21 N in a 5-s period were recorded by the sensors, as shown in Fig. 6c. The responses generally match well without the critical error that is checked with 0.53 N of the average error. The performance of the fingertip force and the sensing capability shows that the robotic hand has significant potential to achieve force control while performing delicate tasks.

The developed hand was used to crush aluminum cans. Supplementary Video 2 and Fig. 6d show the crushing of the can through power grasping using the hand. At this time, the measured maximum force at each finger was 25 N. As shown in Fig. 6e, the egg can be safely grasped by the hand. Finally, experiments involving grasping of various shapes of objects with various shapes were conducted with Yale-CMU-Berkeley

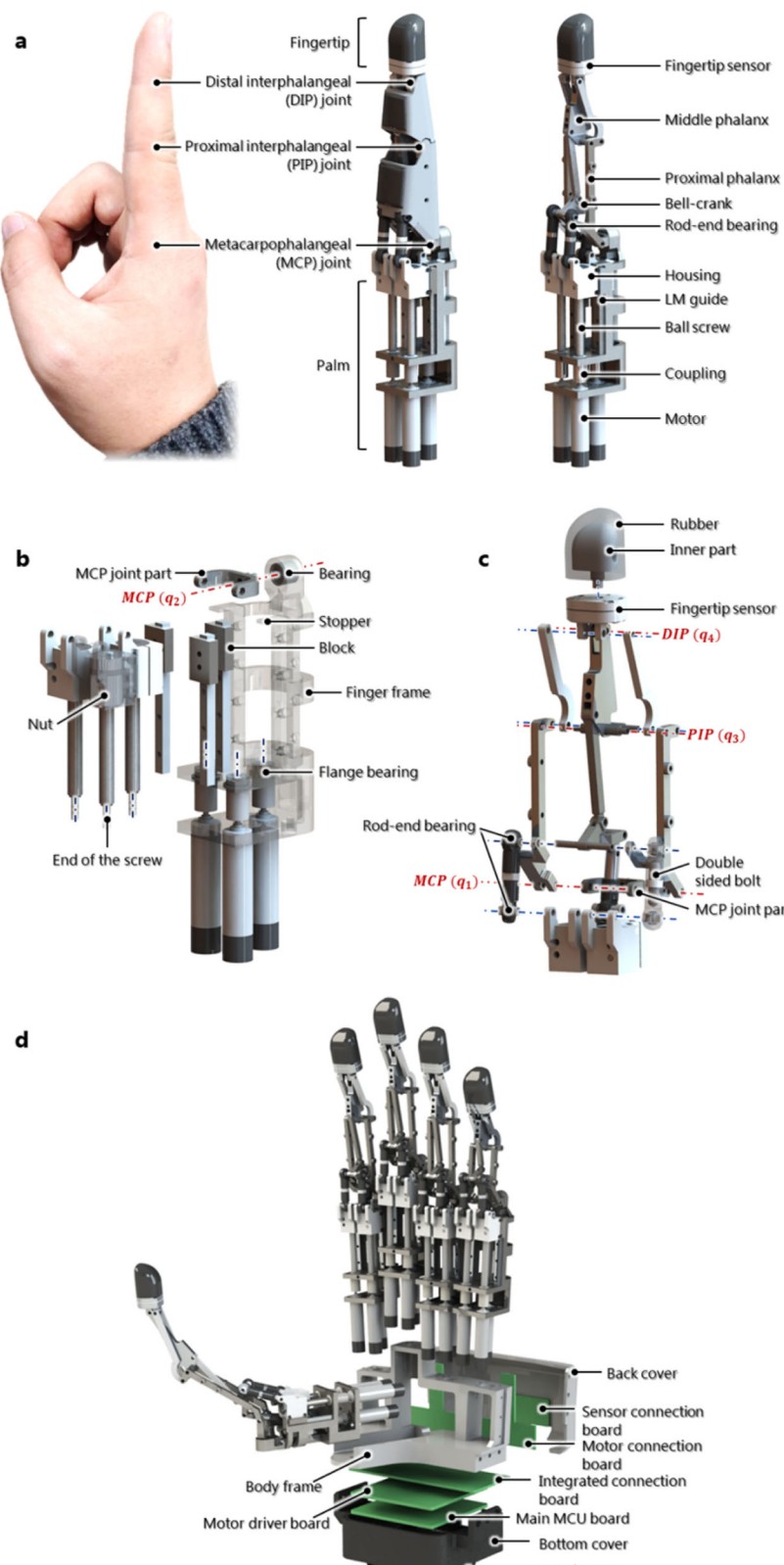

**Fig. 4 Structure of the robotic finger designed based on the proposed mechanism. a** Configuration of the robotic finger. **b** Exploded view of the palm side of the finger. **c** Exploded view of the upper portion of the palm side. **d** Configuration of the ILDA hand. The hand includes the five fingers, three connection boards, a motor driver board, and a main MCU board.

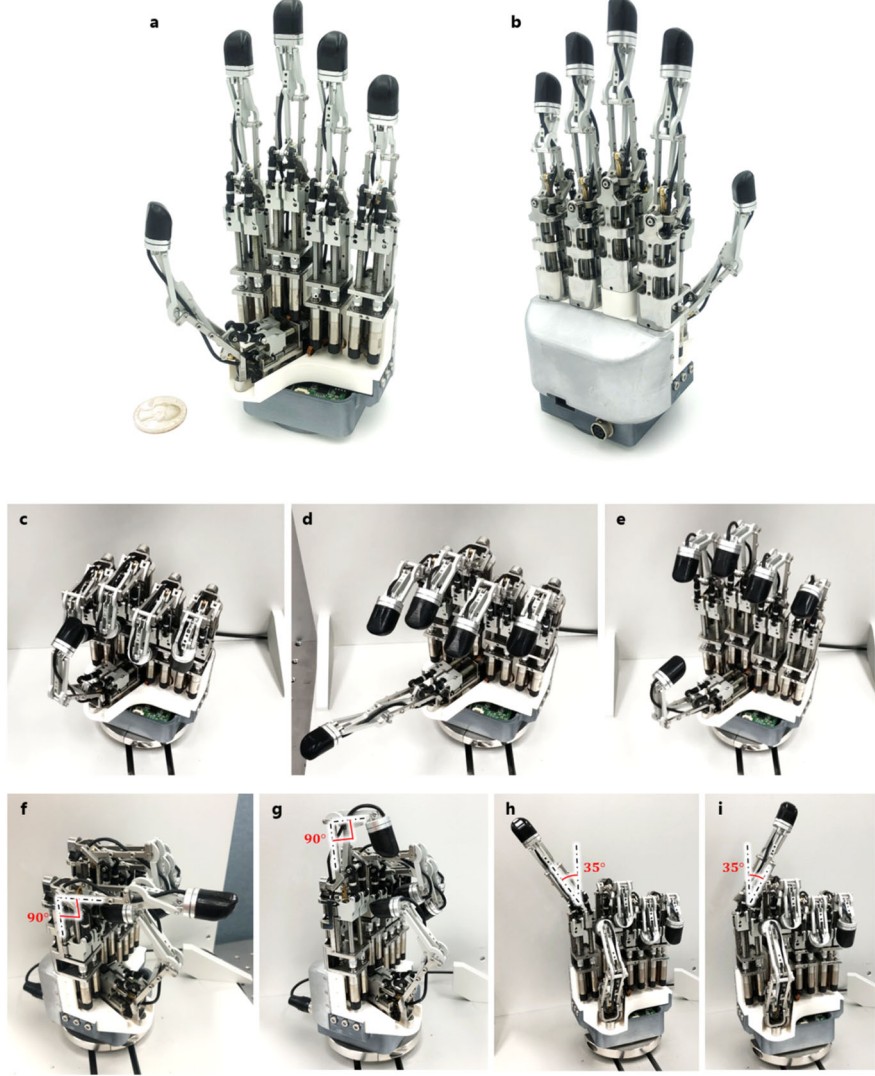

**Fig. 5 Manufactured ILDA hand. a** Front view. **b** Rear view. **c–i** Motion tests of the developed ILDA hand. **c** Fist of the hand. **d, e** Motions of the five fingers of the hand. **f–i** Maximum 3-DOF motions of the finger.

**Table 1 Technical specifications of the developed ILDA hand.**

| Dimensions | Maximum hand length | 218 mm |
|---|---|---|
| | Overall length | 261 mm |
| Weight | 1.1 kg | |
| Active DOF | 15 DOF/20 joints | |
| Fingertip force | Stretched pose | 28 N |
| | Bent pose | 34 N |
| Payload | 18 kg | |
| Joint speed | 53, 103, 81°/s ($q_1$, $q_2$, $q_3$) | |
| Tactile sensing | Force resolution | 62 mN |
| | Force range | ±35 N |
| Power supply | DC15 V | |
| Maximum current | 2 A | |
| Communication | CAN | |

(YCB) objects, and the details are shown in Supplementary Fig. 6 and Video 3. Several experiments were conducted to verify the reliability and robustness of the developed hand, such as test for long-time operation, test to evaluate the ability to grasp for a long time, repeatability test, high payload test, and heating/current test as shown in Fig. 7. This is considered

to be important for the actual use of the hand. Detailed analysis of the experimental results is provided in Supplementary Text 5.

**Manipulation tests with ILDA hand.** To confirm the possibility of tool manipulation using the hand, experiments were conducted by attaching the hand to a commercial robotic manipulator. A paper cutting experiment was performed as cutting with scissors is a task requiring high dexterity in everyday life (Fig. 8a). Further, a small object was moved with tweezers to verify the tactile sensing performance of the hand while the hand delicately grasped the tool (Fig. 8b).

During the paper cutting experiment with scissors, the grasping motion was generated using the control strategy explained in the Methods that the robot hand did not warp when holding the scissors. Here, the scissors used were not customized for the hand, but were general scissors used in everyday life. The robotic manipulator was positioned so that the scissors held by the robotic hand were perpendicular to the paper, and in-hand manipulation was performed to provide adequate workspace for the scissors while maintaining the grasping stability. The hand successfully cut the paper through organic control with the forward movement of the manipulator as the paper moved

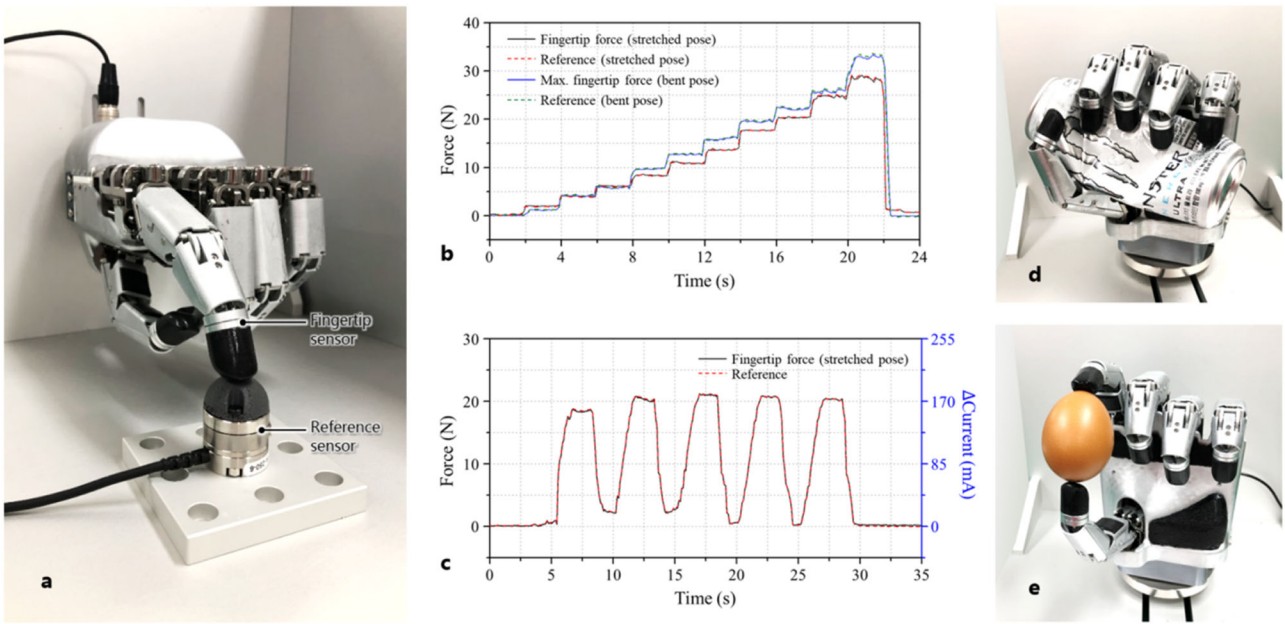

**Fig. 6 Performance tests of the hand. a** Experimental setup to evaluate fingertip force, control ability, and force-sensing performance of the developed hand. **b** Measured force data when the force applied by the fingers was sequentially increased stepwise. **c** Measured force data when applying the force sinusoidally. **d** Robot hand squeezing aluminum can with strong gripping force. **e** Robot hand holding an egg with delicate grasping.

**Fig. 7 Reliability tests of the hand. a** Measured contact force and temperature of the hand during a long-time operation of a robotic finger. **b** Measured temperature and current of the hand when grasping a ball by the hand. **c, d** Measured force data when repeating full stretched pose and bent pose of the finger with 5-s and 1-s contacts, respectively. **e, f** Contact locations on the reference sensor of the finger with 5-s and 1-s contacts. **g, h** Holding and lifting the 18-kg dumbbells for a payload test.

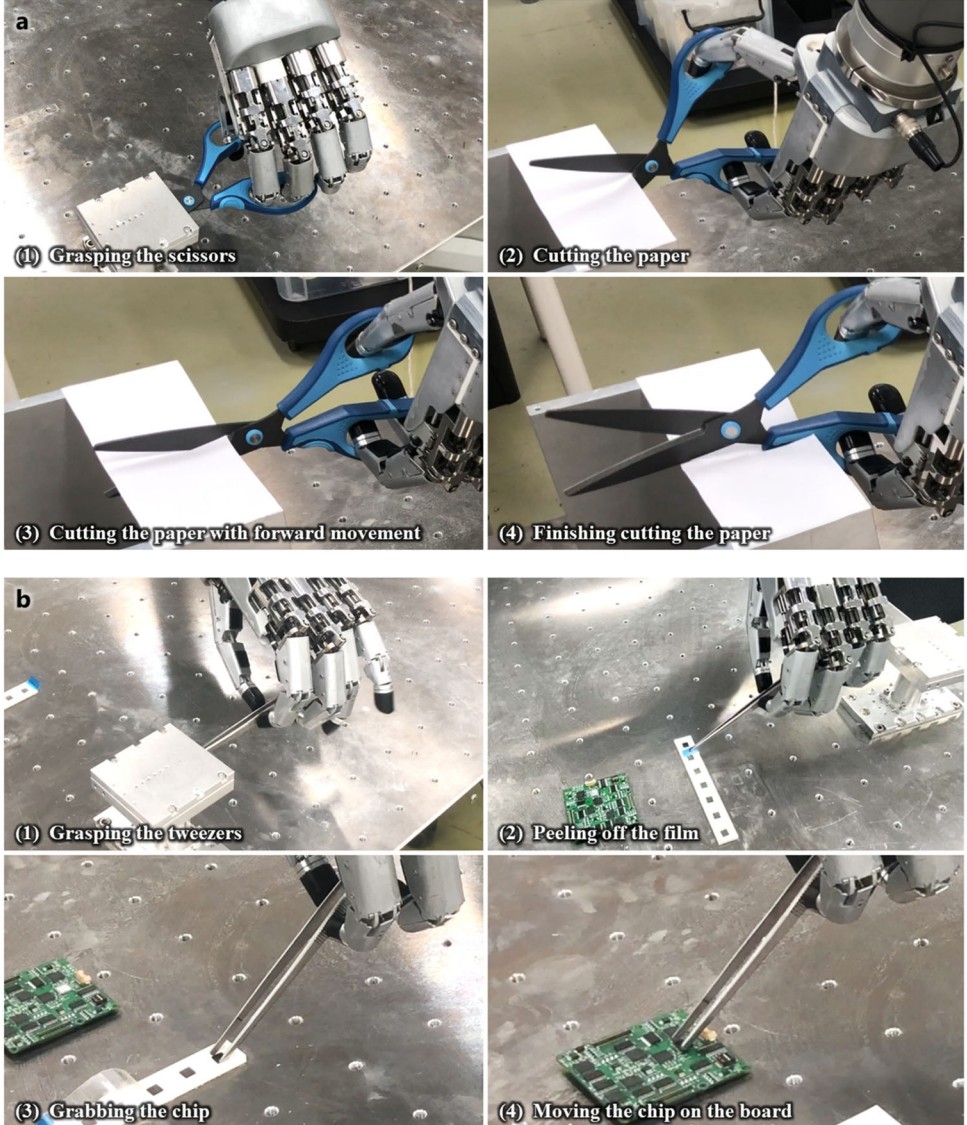

**Fig. 8 Tool manipulation tests with the hand fixed to the robotic arm. a** Experiment where the hand cuts paper by holding and manipulating scissors. **b** Experiment where the robot grabs tweezers and manipulates the tips of the tweezers to grab and move a small object.

simultaneously while being cut with the scissors. A combination of three-finger organic movements was used to create movements for in-hand manipulation, and the image during this experiment is presented in Supplementary Video 4.

Next, an experiment involving holding and moving small objects with tweezers was conducted. First, the hand grasped the tweezers, and the position of the tweezers was changed to an easy-to-hold direction corresponding to the small object, which is a flat square chip of a thickness of 0.9 mm, and a length of 6 mm through in-hand motions. The tweezers used in this experiment are commonly used in homes and laboratories. The manipulator moved such that the tips of the tweezers could hold the small chip, and the hand performed the grasping motion such that the tweezers peeled out the chip's cover and grasped the chip. During grasping, we could determine how much force was applied at the fingertips through the fingertip sensor. Next, the object was moved to another location and the tweezers were released to complete the operation. The force data obtained at this time are shown in Supplementary Text 3. The experiment is demonstrated in Supplementary Video 5. The results proved that this compact robotic hand could be easily attached to a commercial robot arm.

Further, it has high dexterity and force-sensing functions and could complete difficult tool operations.

## Discussion

A dexterous anthropomorphic robotic hand was designed based on a linkage-driven mechanism. The mechanism of the developed robotic hand ensures that the original advantages of linkage-driven mechanisms such as bidirectional control of joints, robustness, and ease of manufacture and maintenance are maintained. At the same time, it ensures active 15-DOF movements with 20 joints, sufficient workspace between the fingers, and high fingertip force (34 N); it also weighs less (1.1 kg), has a compact size (maximum length: 218 mm), and provides space for sensor integration. It can be easily attached to existing commercial robotic arms or those under development because all parts are integrated in the hand without additional parts such as forearms. Therefore, the key advantage is that the hand exhibits high performance and the part configuration is combined with the hand itself.

There are many controller-embedded robotic hands. However, most of these hands have very low DOF. It is not difficult to

realize robotic hands in the form of an integrated hand because the number of drivers required is small. In addition, most of the integrated hands that implement a high DOF have weak gripping force (or fingertip force) and payload. Therefore, in order to ensure a high DOF as well as strong gripping force and payload, tendon-driven mechanism-based robotic hands that have a driving part shaped like a forearm have been developed as dexterous robotic hands.

The grasping ability of the ILDA was confirmed when it squeezed the aluminum can and delicately grasped the egg. In addition, the hand could perform different types of grasping according to the shapes of various objects. Scissors and tweezers are used to determine the possibility of manipulating tools used in everyday life. Although it was difficult to accurately quantify the effectiveness of the hand in manipulating tools using scissors, we performed a combination of movements using many DOF of the hand and through bidirectional control of the joints. As shown in the video, the manipulation movements performed using scissors were very similar to that of humans. It is easier to manipulate tweezers compared to scissors as the hand movement is slightly simpler. However, it is not easy to hold the tweezers, turn them in the desired direction, and maintain the grip while holding the object. We were able to determine whether the object was grasped correctly by observing the fluctuation in the measured force while holding a very small object, indicating high potential for easy force control when using the robot hand in future applications.

The development of a dexterous anthropomorphic hand with a high DOF, which is still an open issue, requires researches from a scientific as well as an engineering perspective. In this study, we tried to maximize the market penetration of robotic hand in various research fields. For this, a robotic hand was developed by combining the scientific issue of developing a mechanism for robotic hand and cutting-edge engineering technology. Thus far, many dexterous robotic hands have been developed, but their universal use has been limited by high cost resulting from complex manufacturing processes and difficulties in maintenance. It is expected that its applicability will spread to actual research fields and the industry through considerations of functionality and cost. As a result, we hope to contribute to the development of the various fields of robotics by implementing the developed robotic hand. We consider that it is very meaningful to use the robotic hand to perform paper cutting with scissors. Further, we expect that the experiment of transferring chips with tweezers will have significant implications for scientific progress. In the future, we will pioneer new research fields by performing tasks that have not been done before in cooperation with other researchers.

In conclusion, the designed ILDA hand that can be easily attached to robotic arms, while simultaneously accommodating various advantageous features, can be used for further research on robotic hands in various fields.

## Methods

### Fabrication of the ILDA hand
A coupling for matching the shaft concentricity of the motor and ball screw was connected between them (Fig. 4b). Flange bearings fixed on the finger frame protect the motors from axial forces. Each end of the three ball screws was placed on the hole of the bearing. A housing part covering the nut of the ball screw was fixed to the block of the LM guide. Each rail of the LM guide was fixed using a bolting connection. The holes in the upper left portion of the housing part were used to fix the rod end with a shaft pin. Because the torque in the ball screw generated by the force from the rod end decreases the driving efficiency of the ball screw, the LM guide was arranged to cancel the torque. Because the LM guide can sustain torque and create linear motion, there is no need to fix the top of the ball screw. Accordingly, the movement of the MCP joint could be secured further, and the assembly of the parts became easier. Because the nuts and the block should not come off the ball screw and the rail, stoppers were constructed on the finger frame. The power transmission part that facilitates three independent linear motions was compactly constructed. In addition, the MCP joint part was fixed using a double-row bearing, enabling the abduction and adduction. The rod end was fixed by a double-sided blot to the housing, and the other end of

the rod was fixed to the other side of the bolt. These connections form the PSS chain. The rod end was pinned to the proximal phalanx, and the inner rod end was fixed using a bolt with a universal joint to form a PSU chain. As illustrated in Fig. 4c, the universal joint was pinned to one side of the bell-crank and the proposed four-bar linkages above. It consists of the pin assembly of the revolute joint in the linkages. In addition, sensor wiring is easy because there are many empty spaces in the structure.

The process used for fabricating the finger, the key element in the hand, is described here. All parts of the finger assembly are shown in Supplementary Fig. 3a. There are not many parts in the fingers of the dexterous hand, and all of them are easy to manufacture. A ball screw (SR0401K, KSS) was machined to erase the flange to connect with the housing. The housing was bolted to the LM guide (LWL3, IKO). The most robust finger frame and proximal phalanx, MCP joint part, and shafts, which may be damaged, were made of SUS303 steel. The rest of the parts were mostly made of aluminum 6061. A small motor (DCX 8 M, Maxon) with a diameter of 8 mm and a gearbox (GPX8, Maxon) with a reduction ratio of 16:1, and an encoder (ENX 8 mag, Maxon) was used. The motor that was used has a no load current of 2.74 mA, a maximum stall current of 0.13 A, and a nominal voltage of 12 V. Therefore, the maximum stall current of the 15 motors is 1.95 A. A silicone material (KE-1300, Shin-Etsu) was used to increase the friction by covering the inner part to create a fingertip shape. The procedure for assembling the parts of the finger is shown in Supplementary Fig. 3b.

### Configuration of the electronic part
The 15 motors are connected to the motor connection board, which is connected to the integrated connection board with five merged flexible PCB-based wires. The integrated connection board and the motor driver board are connected through pin headers and sockets. The motor driver board is connected to the main MCU board in the same manner. In addition, the five fingertip sensors are connected to the sensor connection board, and the integrated wiring is connected to the main MCU board. The configuration is completed with a four-pin connector installed on the back of the bottom cover. The connection board is capable of controlling communication in all the motors and fingertip sensors at the same time and provides a compact and highly complete wiring configuration for the robotic hand. The electronic part has a compact size in the lower part of the palm, and the communication and power sources are completed with a four-pin connector.

### Experimental setups
For communication between the motors and the force sensors, eight dual full-bridge motor drivers (A3909, Allegro Microsystems) that drive fifteen DC motors and eight slave MCU chips (STM32F411, STMicroelectronics) that transmit control commands to the driver were installed on the motor driver board. The main MCU board comprised a main MCU chip (STM32F407, STMicroelectronics) for communicating with the slave MCU chips through the SPI bus. The fingertip sensors communicated with the main MCU via I$^2$C communication. The main MCU was configured to communicate with the CAN bus on the top controller desktop.

The experimental environment comprised the developed ILDA hand and a commercial robotic manipulator (UR-5, Universal Robots) with a controller, and a desktop as the top controller. The manipulator has a payload of 5 kg and is controlled by a controller that synchronizes the hand using digital input/output signals. To transmit commands to the hand and gather data from the encoder and the sensors, MFC programming was performed using Visual Studio 2019 on the desktop. In addition, to monitor the motion of the hand and the measured contact forces, OpenGL was used as the cross-platform programming interface with MFC programming to render 3D vector graphics.

### Control strategy
The robotic finger has several DOF, and it is difficult to control all the joints intuitively. Hand synergy refers to aligning joint angle trajectories depending on the type of grasping[46,51]. The overall joint motions depending on the type of grasping can be represented by some principal components (PCs). The joint angle representation using synergy vectors is given by $q = S\sigma$ where $q$ is the joint angle of the robotic hand, $S$ is the synergy vector, and $\sigma$ is the coefficient of the synergy vector. Synergy vectors are PCs of the types of grasping that humans usually perform. General grasping types in manufacturing were proposed by Cutkosky[52]. A total of 33 types of grasps that can represent all human hand motions were proposed by Feix et al.[45]. Strategies to grasp and manipulate various objects were used through hand synergies. First, information regarding the synergies was collected for the given objects. The synergy information includes the suitable types of grasping and the coefficient of the synergies, which is proportional to the size of the objects or the amount of motion. The 14 essential grasp types for grasping various objects were chosen from Feix's grasp taxonomy. The joint angle vector in every grasp type is predicted by the visual simulator of the ILDA hand. The hand synergy vector is calculated by analyzing the PCs of the change of joint angle vectors in different sizes of objects and motions from the simulator. Then, each action of the hand was defined using the synergy information of the objects. With the command for the actions, the trajectory of the joints and the desired motor displacement are calculated using the synergy information. Therefore, the motors using the position controller are controlled, and the robotic finger moves to grasp objects. To grasp and manipulate the tools, grasping and manipulating

strategies were established. Human hand's actions are analyzed for specific manipulation tasks. We separated the actions into independent components, which could be represented by one PC of the joint angle vector. The hand synergy vectors for the manipulation task are calculated by the PCs from the motion teaching of each separated action. We teach each action of the hand and the manipulator and create a scenario for a given task. The robot environment was controlled in the task scenario

**Tactile sensing capability**. The delicate functions of the robotic hand require tactile sensing capabilities. Several types of external forces are applied to the fingertips of the hand, especially during grasping or manipulating objects and tools of various shapes[53]. Therefore, a six-axis F/T sensor was installed to measure the various forces acting on the robotic fingertips. The F/T sensor can measure three orthogonal forces ($f_x$, $f_y$, and $f_z$) and three orthogonal torques ($t_x$, $t_y$, and $t_z$), which can be utilized to derive the location and triaxial contact force applied on the fingertip[54–56]. This miniature sensor used for measuring the forces on the fingertips was developed, and detailed information is provided by Kim[50].

To provide tactile sensing capabilities to the robotic hand, the fingertip was designed as shown in Supplementary Fig. 5. The rigid inner part is integrated on the sensor to apply the correct external force without distortion. The surface is covered by a rubber material that can increase the friction force required for easy grasping of objects. This is a suitable robot fingertip sensor because it measures different types of forces. The wiring of the sensors in the linkage structure ensured no interference with the hand motions, and algorithms were used to determine the contact force parameters, including the contact location and triaxial force applied on the fingertip. A detailed description of the algorithm for force sensing is included in Supplementary Text 4.

## Data availability
All data needed to support the conclusions are available from the corresponding author upon reasonable request.

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

## Acknowledgements
This work was supported by the new faculty research fund of Ajou University. This work was also supported by the National Research Council of Science & Technology (NST, Korea Government) under Grant NK233C. This work was also supported by the new faculty research fund of Ajou University. U.K. would like to thank J. Chung for her encouragement to continue meaningful research.

## Author contributions
U.K. developed the robotic hand including its design, fabrication, and control, as well as the experiments. D.J applied the control strategy to perform the experiments, and wrote the manuscript. H.J. assisted in designing the robotic hand and wrote the manuscript. J.P. developed the control software of the robotic hand. H.J assisted in designing the robotic hand. J.C. applied the control strategy and edited the manuscript. H.R.C. directed the study and edited the manuscript. H.D. and C.P. were responsible for the overall research direction and objectives.

## Competing interests
U.K., H.J., H.D., J.P., and C.P. are inventors in the patent application (Kor. 10-2019-0167204) submitted by Korea Institute of Machinery and Materials that covers the mechanism of the robotic finger. The other authors declare that they have no competing interests.
