## [Peer Review File · Nature Communications]

Reviewers' Comments:

Reviewer #1:

Remarks to the Author:

The submitted version looks like a technical report on the robotic hand development procedures. The proposed robotic hand conducts many hard tasks, but it is difficult to find out what the scientific findings are, except the well-disciplined engineering works.

In addition, ambiguous and very subjective terminologies are widely used in the manuscript. For example,

In abstract,

"Additional large parts are required to control the hands and they cannot be integrated into existing robotic arms, thus limiting their applicability."

=> The reviewer cannot agree with the above argument. We have already many controller-embedded robotic hand models as commercialized versions.

In the introduction,

"Based on the above analysis, we concluded that it is important to have the following advantageous features in a robotic hand: high dexterity, high fingertip force, ease of control, robustness, low cost owing to easy manufacturing, ease of maintenance, compactness, low weight, and compact wiring."

=> Definitions are very unclear and ambiguous. For example, what is the definition of high dexterity? High fingertip force? Ease of control? Compactness? Low weight? Especially, compact wiring?

The followings are my technical comments.

In the results,

"To overcome the aforementioned shortcomings and include advantageous features, the mechanism of the robotic finger was designed according to the schematics in Fig. 2."

=> For this purpose, workspace analysis is in advance required, but the reviewer could not find it in the manuscript.

"The mechanism for driving the PIP joint is to implement the motion of the MCP joint and independent movement in a narrow space. Most linkage-driven mechanisms are composed of solid links, so it is difficult to realize the independent movement."

=> The reviewer could not understand it. It looks like a personal opinion by the authors.

"Here, the bell-crank part enables efficient transmission of the driving force."

=> Efficiency is well-defined terminology having the mathematical expression. Prior to arguing the above, the authors should suggest how efficient compared to the existing other transmission methods.

"At this time, d3 slightly decoupled the motion of the PIP."

=> The term "slightly" is also ambiguous

"At this time, the PSU chain can cope without affecting the PIP joint."

=> The reviewer could not understand it

"To realize the proposed linkage-driven mechanism, we mainly considered the following factors:

(1) selection of small parts that can play the role of each joint, (2) part placement and configuration to achieve the desired DOF motion and drive angles, (3) efficient power transmission structure, and (4) ease of manufacture and assembly."

=> Too subjective statements.

"The rod ends serve several important functions. First, because it is possible to continuously rotate in the axial direction without the cover of the rod end, the rod end enables 90° of movement during flexion and extension of the MCP joint, and ±35° of movement is possible during abduction and adduction owing to the characteristics of the rod end."

=> These characteristics are dependent on the mechanism dimension and configuration.

Reviewer #2:

Remarks to the Author:

The paper presents the design and performance of a linkage driven robotic hand. The hand is claimed to be dexterous, powerful, intrinsic (both in terms of actuation unit, electronic boards and sensing), and anthropomorphic. Supporting videos show the motion abilities of the fingers and some impressive manipulation actions in challenging tasks. The claims are novel, considering that the development of a compact anthropomorphic hand is still an open issue. The paper is clearly structured and, in my opinion, very interesting even if targeting a narrow field (i.e. robotics, specifically design and control of artificial hands, partially prosthetics). The transmission design is smart since it reduces the complexity of the hand, still allowing for a very dexterous motion and producing large force at the fingertip in different configurations.

In terms of dexterity there are however similar hands already developed:

- The hand within the Modular Prosthetic Limb (MPL)

[<https://www.youtube.com/watch?v=DjzA9b9T3d8>

http://mindtrans.narod.ru/pdfs/Modular_Prosthetic_Limb.pdf], not cited in the paper, and with figures that are similar or better.

- The Shadow Dexterous Hand is similar, but bulky and heavy if compared with the hand here presented (https://www.shadowrobot.com/wp-content/uploads/shadow_dexterous_hand_technical_specification_E_20190221.pdf)

The robustness of the hand is not adequately assessed, in my opinion. While I agree with the authors that the design proposed is very simple, still allowing dexterous motions, the paper does not provide any evidence that the hand is reliable in the long run and can withstand large external loads (impacts for instance) or maintain the grasp for a long time (heating issues since it is back drivable. In this respect it may be an option to also mention the "nominal" force at the fingertip).

The integration with the sensors is very nice, even if the intermediate areas of the fingers are not covered. This limits the sensory information that can be obtained by the hand when power grasps are performed.

I was impressed by the relatively low power consumption (2A at 15V), but I am not sure if it refers to a single finger or if it is the peak of hand when all the fingers exert the maximum force.

Within the paper, the authors refer to "integrated linkage-drive". What does it mean integrated linkage-drive? Is it equal to the "linkage-driven mechanism/configuration" or add something more?

"...Most linkage-driven mechanisms are composed of solid links, so it is difficult to realize the independent movement..."

Not clear here the meaning of "solid". Is it the hand presented here composed of "non solid" linkages? Is it stiff instead of "solid"

"...The three joints, fingertips, and palm of the robotic hand are similar to those of the human hand..."

Why are you not referring to human kinematic architecture and percentile? The hand is roughly two times larger than the 50th of the male human hand in length and weight ("...maximum length of 218 mm and weight of 1.1 kg...").

"...it is very important to use parts with high stiffness that can sustain the force without becoming damaged..."

Strength instead of stiffness? Low stiffness does not mean low strength in most of the case. Let us

think for instance to a compression spring: the low stiffness is due to the long path the stress follows from the load application point to the frame (while the stress is maintained under the strength limit).

Sometimes the authors have used the word configuration instead of mechanism. This is, in my opinion, confusing.

Reviewer #3:

None

Response Letter

Dear Reviewers,

We wish to re-submit a revised version of our manuscript to NATURE COMMUNICATIONS (manuscript ID: NCOMMS-20-36558-T) titled “Integrated Linkage-Driven Dexterous Anthropomorphic Robotic Hand.”

We thank you and the reviewers for your thoughtful suggestions and insights. The manuscript has benefited from these insightful suggestions. We look forward to working with you and the reviewers to move this manuscript closer to publication in NATURE COMMUNICATIONS.

This document contains our responses to the reviewers’ comments and suggestions on how to improve the manuscript. Each reviewer is addressed individually, with the reviewer’s comments in boldface and our answers in italics. We hope that our responses and the revised manuscript satisfactorily address the reviewers’ concerns.

Thank you for your consideration. We look forward to hearing from you.

REVIEWER COMMENTS

Reviewer: 1

Comments to the Author

Q1. The submitted version looks like a technical report on the robotic hand development procedures. The proposed robotic hand conducts many hard tasks, but it is difficult to find out what the scientific findings are, except the well-disciplined engineering works.

Thank you for highlighting the shortcomings of the paper. Based on your comments, we have clarified the purpose of our study and have corrected several vague expressions in the text. Furthermore, we have added important information related to workspace analysis and bell-crank design.

The development of a dexterous anthropomorphic hand with a high degree of freedom, which is still an open issue, requires research from a scientific finding point of view as well as an engineering perspective. We attempted to maximize the ripple effect of robotic hands in various research fields. Therefore, we developed the current robotic hand by combining the scientific issue of establishing a new mechanism for robotic hands and cutting-edge engineering technology.

To emphasize the difference between our robotic hand and previously developed hands, a quantitative comparison was performed (as detailed in the response to Q2). Consequently, it is evident that there is no robotic hand that matches the performance indicators of our developed hand such as a fingertip force of 34 N, payload of 18 kg, and weight of 1.1 kg with the same active 3 DOFs as a human finger without additional parts like a forearm. In particular, based on the linkage-driven mechanism, none of the previously developed hands allowed active 3 DOF movement in one finger.

To reinforce the scientific findings of our study, as suggested by the reviewers, workspace analysis (QT1: technical comment 1), comparative analysis with human hand, and driving efficiency analysis (QT3) were performed. Furthermore, to emphasize the engineering work of the study, we added several experiments to verify the reliability and the robustness of the developed hand, such as long-time operation, ability to grasp for a long time, repeatability test, high payload test, and heating/current test. This is considered to be crucial for the actual use of the hand as presented Fig. 7 and Supplementary Videos 1 and 2.

In addition, we considered the ripple effect of the hand as follows. Until now, many dexterous robotic hands have been developed, but their widespread use has been limited by high price due to complex manufacturing processes and difficulties in maintenance. The use of such hands in actual research fields and industrial sites is expected to be achieved through function and price considerations. Therefore, we hope that developed robotic hand can contribute to the development of the various fields of robotics. We believe that the paper cutting experiment with scissors is meaningful, and that the experiment of transferring chips with tweezers will have contribute to scientific progress. In the future, the developed robotic hand is expected to achieve considerable market penetration through broadcasting media. In addition, we aim to conduct pioneering work in new research fields by performing tasks that have not been conducted before in cooperation with other researchers. The corresponding contents have been added in the Discussion and Conclusion Sections.

Q2. In addition, ambiguous and very subjective terminologies are widely used in the manuscript. For example, In abstract, “Additional large parts are required to control the hands and they cannot be integrated into existing robotic arms, thus limiting their applicability.”

=> The reviewer cannot agree with the above argument. We have already many controller-embedded robotic hand models as commercialized versions.

As the reviewer rightly pointed out, expressions that are too subjective or vague may confuse the reader. Accordingly, we have modified these expressions to avoid any misunderstanding.

We focused on developing a robotic hand that has three DOFs per finger for its dexterous motion by integrating all actuators and controllers. As the reviewer mentioned, there are many controller-embedded robotic hands. However, most of these hands have very low DOF. It is not difficult to realize robotic hands in the form of an integrated hand because the number of drivers required is small. In addition, most of the integrated hands that implement a high DOF have weak gripping force (or fingertip force) and payload. Therefore, in order to ensure a high DOF as well as strong gripping force and payload, tendon-driven mechanism-based robotic hands that have a driving part shaped like a forearm have been developed as dexterous robotic hands^{16,17}.

Thus, we changed and added an explanation to clarify the purpose of the study in the Discussion and Conclusion. Human fingers have 3 active DOFs, and the dexterous robotic hands we refer to aim to have 3 active DOFs per finger. The definitions of various

terms, including dexterity, have been provided in the response to Q3.

In addition, based on the reviewer's comments, we closely compared the proposed multi-fingered robotic hand with those being studied or commercialized worldwide. Therefore, a specification comparison table (Supplementary Table 1). In this table, we mainly compare dexterous robotic hands with more than 3 DOFs per finger, similar to the aim of the present study. Supplementary Table 1 presents detailed specifications of the hands, such as the number of active DOFs, hand size, weight, presence or absence of forearm, fingertip force, payload, and driven type. In summary, there is no hand that has a 3 active DOFs per finger without an additional part (forearm) and has a higher fingertip force and payload than the developed hand. In addition, the developed hand was found to be excellent in terms of size and weight.

Q3. In the introduction, “Based on the above analysis, we concluded that it is important to have the following advantageous features in a robotic hand: high dexterity, high fingertip force, ease of control, robustness, low cost owing to easy manufacturing, ease of maintenance, compactness, low weight, and compact wiring.”

=> Definitions are very unclear and ambiguous. For example, what is the definition of high dexterity? High fingertip force? Ease of control? Compactness? Low weight? Especially, compact wiring?

As the reviewer highlighted, there are many unclear definitions in this manuscript. We added explanations to clarify the mentioned features as follows. In addition, we removed ambiguous expressions.

Dexterity – Because the active motion of a human finger has three DOFs³⁴, we define high dexterity as that with more than three DOFs per finger. A human-level DOF is required to imitate movements mainly used by humans or to manipulate tools.

Fingertip force – The fingertip force of a human (based on the index finger) is defined as the high fingertip force. In previous studies³⁵, the human fingertip force has been reported as 27.9 N. As presented in Supplementary Table 1, it is very meaningful to achieve high fingertip forces and payloads because these are low for most of the dexterous robotic hands.

Controllability – Many types of robotic hands use compliant parts, such as springs,

to efficiently transmit the driving force. It is difficult to completely model the characteristics of the mechanism using these compliant parts. There are a relatively large number of disturbance factors with a level of uncertainty in terms of control, compared to those in the case of mechanisms without compliant parts. Therefore, special control methods to solve the disturbance factors are required^{36,37}. Among the currently developed robotic hands, there are many under-actuated robotic hands that use the adaptive grasping function. Because their structures may cause position errors owing to their compliance when force is applied in a specific situation (e.g., pinch grasp), special control methods are required, or the use is limited to a specific situation. Therefore, if all movements are implemented such that these are dependent on links, the occurrence of such errors is low, and the control is intuitive and simple.

Maintenance – The proposed mechanism provides easier maintenance than that of the tendon-driven mechanism. The hands based on the tendon-driven mechanism are rather difficult to assemble, and the problem of tendon-wire loosening often occurs with longer periods of use. Reassembly for its maintenance is not simple. Difficulties are encountered in the maintenance owing to an increase in cost. In contrast, the proposed mechanism is composed of a linkage structure, simplifying the assembly and maintenance.

Compactness – It describes the need to implement the shape of a hand as all parts (linkage structure, motors, sensors, and controller) are compactly integrated.

Weight – The robotic hand's weight directly affects the performance of the robotic arm when the hand is assembled with the arm. Therefore, the lower the weight, the more advantageous it is to operate the robotic hand-arm system. Weight was judged to be included in compactness, and thus it was removed from the text.

Compact wiring – Compact wiring implies that the wires of the force sensors or motors can be packaged and manufactured compactly inside the hand without being exposed to the outside environment. However, as the reviewer said, it is difficult to define this clearly, and thus it was removed from the text to avoid any confusion.

To validate the performance of the developed hand based on the features described above, the performance indicators of the previously developed dexterous robotic hands are summarized in Supplementary Table 1. The contents are added in Supplementary Text 1.

THE FOLLOWINGS ARE MY TECHNICAL COMMENTS.

QT1. In the results, “To overcome the aforementioned shortcomings and include advantageous features, the mechanism of the robotic finger was designed according to the schematics in Fig. 2.”

=> For this purpose, workspace analysis is in advance required, but the reviewer could not find it in the manuscript.

As the reviewer mentioned, the workspace analysis of the proposed mechanism is important. Therefore, we performed a workspace analysis of the proposed mechanism, and compared it with the workspace of a human finger.

To check the workspace of the designed robotic finger, the reachable workspace of the robotic finger was analyzed as shown in Fig. 3e–g. Fig. 3e shows the 3D view of the reachable workspace composed of points that can be reached by the end point of the finger. Here, d_r is the distance between the end point of the finger and the fixed MCP joint of the finger, which is taken as the origin. The farthest points are expressed in red. In this case, both PIP and DIP of the finger at this time are stretched poses. Fig. 3f shows the side view of the workspaces of the proposed robotic finger and human finger. As it is easy to adjust the length of the three phalanges of the robot finger, from a design perspective, the length of the phalanges of the human finger was analyzed under the same conditions as those of the robotic finger, for a close comparison. Fig. 3f shows the side view of the workspaces of the robotic finger and human finger. Here, $T_1 - T_5$, T_a , and T_{ab} indicate the trajectories of the fingers. The trajectory from T_1 to T_2 shows that the MCP joint moves from full extension (0°) to full flexion (90°) with the stretched pose of the PIP and DIP. The trajectory from T_2 to T_4 is generated when the PIP and DIP are moved while fixing the MCP joint in full flexion. T_3 represents the lowest point on the z-axis in this trajectory. T_5 is reached when the MCP joint is fully extended while maintaining full flexion of the PIP and DIP joints. The trajectory of a human finger is indicated by the dotted line. Fig. 3f shows a workspace similar to that of the two fingers.

The abduction/adduction motion is maximum (30°) when the MCP joint is in full extension and minimum (0°) when the MCP joint is in full flexion⁴⁷. For this reason, the workspace volume under the MCP joint is relatively small as shown in Fig. 3g, and a part of its lower right corner becomes sharp as shown in Fig. 3e, g.

The shape of the workspace at the bottom of the robotic finger is similar to that of

the human finger. The reason is that when the MCP joint is in full flexion, the abduction/adduction movement is limited because the proximal phalanges contact with the top of the two ball screws as shown in Fig. 2e. T_a and T_{ab} are the locations of the maximum abduction/adduction movement during full extension of each joint of the human and robot fingers, respectively. The robotic hand is capable of 35° movement to perform abduction/adduction. As a result, the workspace shapes of the robotic finger and human finger are very similar, and the volume of each workspace was found to be 188740 and 119800 mm³, respectively.

Additionally, detailed comparisons of the robotic finger and human finger are shown in in Supplementary Text 7. The survey was conducted based on the average male hand^{48,49}. Detailed analysis results of this comparison are presented in Supplementary Table 2.

QT2. “The mechanism for driving the PIP joint is to implement the motion of the MCP joint and independent movement in a narrow space. Most linkage-driven mechanisms are composed of solid links, so it is difficult to realize the independent movement.”

=> The reviewer could not understand it. It looks like a personal opinion by the authors.

We agree that the point made may be misleading. Therefore, the sentences were revised for greater clarity as follows.

→ *To implement a single DOF of the PIP joint, the PIP joint must be driven independently of the MCP joint. Most linkage-driven robotic fingers have realized only 1- or 2-DOF movements in which the two joints are subordinated. As a result, a linkage-driven robotic finger with three DOF has not been investigated so far, as shown in Supplementary Table 1.*

Additionally, to drive the PIP joint, power should be transmitted to the PIP joint through the MCP joint, and it is difficult to transmit the power by the movement of the MCP joint. In the case of the tendon-driven mechanism, the tendons can be easily connected to the two joints and operated independently of the joints due to its flexibility. Further details can be found in previous publications³⁷. Because links are rigid, it is difficult to solve this problem. For this reason, thus far, a 3-DOF robot finger using a linkage-driven mechanism has not been developed. Therefore, we developed a new robotic finger capable of 3-DOF movement using the proposed new linkage-driven mechanism.

QT3. “Here, the bell-crank part enables efficient transmission of the driving force.”

=> Efficiency is well-defined terminology having the mathematical expression. Prior to arguing the above, the authors should suggest how efficient compared to the existing other transmission methods.

Currently, a linkage-driven mechanism that can be used to implement an active 1-DOF PIP joint in robotic hands has not been investigated. Therefore, a comparison with other existing transmission methods under the same conditions is difficult. However, as the reviewer highlighted, there is insufficient information about the design of the bell-crank. Therefore, the transmission efficiency of the driving force according to the design of the bell crank was analyzed.

A bell crank is responsible for changing motion through its angle in a limited space. According to the characteristics of the linkage-driven mechanism, the length of a moment arm is changed by the joint angle. Therefore, the magnitude of the torque generated when a constant force in one direction is applied to the end of the moment arm varies according to the position of the joint. When the link structure becomes more complex, such as a structure connected from d_3 to the PIP joint in Fig. 3 b and c, the torque distortion becomes more severe. Therefore, we used the bell-crank mechanism to compensate for the torque distortion and obtain the largest torque in the desired position of the PIP joint.

The torque transmission rate varies according to the joint position owing to the design of the bell crank. Specifically, among the various design parameters of the bell crank, adjusting the included angle (β_1) of the bell crank in terms of its design is easy. Therefore, we analyzed the efficient torque transmission structure using the bell crank as the design parameter.

The precise point at which the joint of a human finger can generate the strongest gripping force is not known. However, previous studies have shown that the gripping force is stronger in the bent pose than in the stretched pose⁴⁴. Therefore, when each joint angle is in an intermediate position (MCP: 45°, PIP: 45°), securing the strongest gripping force is effective. When manipulating or gripping various objects, the most used joint angle is the intermediate position⁴⁵. Hand synergies constituting the gripping action are also mainly distributed around the intermediate position⁴⁶. Therefore, we analyzed the bell-crank design to secure the highest torque in the intermediate position.

To calculate the transmission efficiency using the included angle of the bell crank

as a variable, we analyzed the free-body diagram of the linkage chains from d_3 to the PIP joint, as presented in Supplementary Fig. 1. A detailed equation derivation for this has been presented in Supplementary Text 3. As a result, a relation of torque (τ_{pip}) generated by the input force (F_{d_3}) to d_3 was derived. As presented in Supplementary Fig. 2 **a-e**, the angle of the MCP joint was analyzed by dividing it into five cases. The graphs show the torque values (τ_{pip}) from 0° to 90° of the PIP joint when an input (F_{d_3}) of 1 N is applied. The torque values are analyzed for three cases of the included angle: 0° (without the bell-crank part), 40° , and 80° . Supplementary Fig. 2 **a** presents the torque data of the MCP joint in the full extension pose, and Supplementary Fig. 2 **e** presents the torque data of the MCP joint in the full flexion pose. When the angle of the MCP and PIP joints was 45° , the included angle of the bell crank that secures the highest torque was confirmed to be 40° , as displayed in Supplementary Fig. 2 **c**. In addition, Supplementary Fig. 2 **f** represents the sum of torques generated in all MCP joint positions ($0^\circ \sim 90^\circ$) according to the PIP angle. The total torque can be derived from $\int_0^{\pi/2} \tau_{pip} q_2 dq_2$. In the graph, when the ball-crank angle is 40° , the highest total torque is secured at a PIP angle of 45° . Further, it is possible to obtain the torque with the smallest fluctuation in the entire section of the PIP joint.

As a result, we found the optimal value of the included angle of the bell crank that can provide the highest gripping force in the 45° bent pose while maintaining high gripping forces in the full extension (stretched) and full flexion poses of all the joints. The contents are presented in Supplementary Text 3.

QT4. "At this time, d_3 slightly decoupled the motion of the PIP."

=> The term "slightly" is also ambiguous.

As the reviewer rightly pointed out, the ambiguity of this expression may confuse the reader. Therefore, the sentence was modified for greater clarity as follows.

→ *At this time, the movements of the MCP do not affect the movement of the PIP joint connected to d_3 .*

QT5. "At this time, the PSU chain can cope without affecting the PIP joint."

=> The reviewer could not understand it

As the reviewer highlighted, the ambiguity of this expression may confuse the reader.

Because this sentence essentially conveyed the same meaning as the sentence presented in QT4, it was removed.

QT6. “To realize the proposed linkage-driven mechanism, we mainly considered the following factors: (1) selection of small parts that can play the role of each joint, (2) part placement and configuration to achieve the desired DOF motion and drive angles, (3) efficient power transmission structure, and (4) ease of manufacture and assembly.”

=> Too subjective statements.

We agree with the reviewer that these statements may appear too subjective. However, the authors believe that some subjective opinion is necessary in the design part. Since the developed mechanism is in a new form, subjective opinion is necessary to design this mechanism. Therefore, additional content has been added for greater detail.

(1) Selection and configuration of parts of appropriate size to achieve the desired DOF motion: To realize the function of the abovementioned kinematic model in the narrow space of a finger shape, it should be properly arranged in the configuration of the model. Therefore, it is important to select small components of a suitable size from a design perspective.

(2) Efficient power transmission structure to minimize the friction between the assembly parts. To achieve high fingertip force, a compact structure is required while minimizing the friction force in the power transmission part.

(3) Ease of manufacture and assembly. To increase the ripple effect of the developed robotic hand, it is also important to evaluate it in terms of cost and maintenance. Therefore, it is important to design a simple and robust robotic hand structure.

The contents of (1) and (2) concerns the functional part of the hand, and (3) is regarding the high ripple effect of the hand. Thus far, many dexterous robotic hands have been developed as summarized in Supplementary Table 1, but their widespread use has been limited by high price due to manufacturing and maintenance difficulties. The use of such hands is expected to be achieved in actual research fields and industrial sites through function and price considerations. As a result, we hope to contribute to the development of the field of robotics by utilizing the developed robotic hand.

QT7. “The rod ends serve several important functions. First, because it is possible to continuously rotate in the axial direction without the cover of the rod end, the rod end enables 90° of movement during flexion and extension of the MCP joint, and ±35° of movement is possible during abduction and adduction owing to the characteristics of the rod end.”

=> These characteristics are dependent on the mechanism dimension and configuration.

We agree with the reviewer that the above sentences are misleading. The movement of the MCP joint is not provided by the rod end itself; it is provided by the entire mechanism, including the rod end. Therefore, the sentence has been modified for greater clarity as follows.

→ *The rod end serves as an important joint in the implementation of this mechanism. First, it is possible to continuously rotate it in the axial direction without using the cover of the rod end. The rod end performs the function of a ball joint without limiting the movement of the MCP joint during the full range (0–90°) flexion and extension of the joint. In general, ball bearings allow a limited range due to the contact between its socket and ball. In addition, during the full range of the abduction and adduction (±35°) of the MCP joint, the rod end can act as the ball joint without impeding the movement.*

Once again, we would like to thank the reviewer for their time and effort in reviewing the previous version of our paper. This has been highly insightful and helpful to us.

Reviewer: 2

Comments to the Author

The paper presents the design and performance of a linkage driven robotic hand. The hand is claimed to be dexterous, powerful, intrinsic (both in terms of actuation unit, electronic boards and sensing), and anthropomorphic. Supporting videos show the motion abilities of the fingers and some impressive manipulation actions in challenging tasks. The claims are novel, considering that the development of a compact anthropomorphic hand is still an open issue. The paper is clearly structured and, in my opinion, very interesting even if targeting a narrow field (i.e. robotics, specifically design and control of artificial hands, partially prosthetics). The transmission design is smart since it reduces the complexity of the hand, still allowing for a very dexterous motion and producing large force at the fingertip in different configurations.

Thank you for reviewing the manuscript and providing a concise and accurate summary of our work as well as high-quality feedback regarding our robotic hand.

The reviewer provided important suggestions regarding the reliability verification for the practical use of the developed hand. We performed several experiments to verify the reliability and robustness of the developed hand, such as long-time operation, grasp for a long time, repeatability test, high payload test, and heating/current test. This is considered to be crucial for the actual use of the hand.

Q1. In terms of dexterity there are however similar hands already developed:

1) The hand within the Modular Prosthetic Limb (MPL)

**[<https://www.youtube.com/watch?v=DjzA9b9T3d8>,
[http://mindtrans.narod.ru/pdfs/Modular Prosthetic Limb.pdf](http://mindtrans.narod.ru/pdfs/Modular_Prosthetic_Limb.pdf)], not cited in the paper, and with figures that are similar or better.**

2) The Shadow Dexterous Hand is similar, but bulky and heavy if compared with the hand here presented ([https://www.shadowrobot.com/wp-content/uploads/shadow dexterous hand technical specification E 20190221.pdf](https://www.shadowrobot.com/wp-content/uploads/shadow_dexterous_hand_technical_specification_E_20190221.pdf))

As the reviewer mentioned, hands with similar performance in terms of dexterity have been developed. We reviewed the contents of the two preliminary studies (1) and 2)) presented by the reviewer. In addition, a table was added to explain the difference between our robotic hand and previously developed ones, and a comparative analysis was performed with various dexterous robotic hands.

We reviewed the references suggested by the reviewer. In particular, the first suggested reference presents a very high-level and well-made dexterous robotic hand. Therefore, the study was discussed as a representative robotic hand in the Introduction Section and was added as an additional reference¹⁵. “MPL v2.0, which was developed by Johns Hopkins APL, shows a high dexterity with active 22 DOF and a compact design integrating actuators and electronics. This hand is capable of human-level natural movement and tactile feedback.” The second study is superior in terms of dexterity. However, unlike our developed robotic hand, an additional large driving unit (forearm) was utilized in this previous study. To consider these issues, several previously developed robotic hands were investigated for comparison with our developed hand.

In addition, we added a table for summarizing the specifications of previously developed dexterous robotic hands and those of the ILDA hand. Supplementary Table 1 presents detailed specifications of the hands such as the number of active DOFs, hand size, weight, presence or absence of forearm, fingertip force, payload, and driven type. In summary, there is no hand that has 3 active DOFs per finger without additional parts (forearm) and has a higher fingertip force and payload than the developed hand. In addition, the developed hand was found to be excellent in terms of size and weight.

Q2. The robustness of the hand is not adequately assessed, in my opinion. While I agree with the authors that the design proposed is very simple, still allowing dexterous motions, the

paper does not provide any evidence that the hand is reliable in the long run and can withstand large external loads (impacts for instance) or maintain the grasp for a long time (heating issues since it is back drivable. In this respect it may be an option to also mention the "nominal" force at the fingertip).

According to the reviewer's comment, we attempted to verify the robustness of the developed robotic hand. The reviewer suggested that this should be done in terms of reliability in the long run, large external loads, long-time grasp, and heating issues. In addition, the applied current was also checked, as discussed in the response to Q4. We verified various aspects related to robustness through several experiments. Here, robustness to large external loads was verified with high payload tests. In fact, since there are few prior studies on reliability verification in most such cases, the authors have developed approaches to this end themselves. We also agree that the parts suggested by the reviewer are very important to verify the reliability for the actual use of the robotic hand.

As the reviewer is aware, the developed hand is based on a link-driven mechanism. The compliance part is not used to eliminate the uncertainty of control, and thus the developed robotic hand will be weak against strong impact. On the other hand, it has the advantage of being robust and easy to maintain. In the event of a strong impact, it is expected that the weakest part of the structure (the part where the link is thin) will break. Therefore, instead of an impact test, we conducted high payload tests. The experimental results were analyzed by conducting several experiments related to the reliability of the hand as follows.

- (1) **Long-time operation of the hand.** We tested whether the hand can stably perform repeated movements for a prolonged period. We repeated free motions of five fingers of the hand for 30 min. It was confirmed that normal operation was performed during the experiment. The long-time operation test is illustrated in Supplementary Video 1.*
- (2) **Constant fingertip force applied by the finger.** To test whether a fingertip force can be continuously applied by the developed robotic finger, an experiment was performed, wherein a fingertip force was applied by a finger for 30 min in the experimental environment displayed in Fig. 6 a. Fig. 7 a shows the measured fingertip force and the temperature of the hand. For accurate measurement, the force was measured by the reference sensor, and the part with the highest temperature on the motor control board was measured every 30 s using a temperature-measuring device (TESTO845, TESTO). The finger was allowed to*

press the sensor for 30 min based on position control with static-force analysis. Experimental results were obtained; the fingertip force measured by the reference sensor continuously decreased from 13.5 N to 11.2 N. One of the reasons for this decrease is thought to be the relaxation effect due to the soft material of the fingertip. For a detailed analysis, a repeatability test was performed. The temperature was measured to be 38.5 °C in the no-load state; it continued to increase over time and converged to 62 °C in 28 min. The current measured during the experiment was constant (0.3 A).

- (3) **Grasping a ball for a prolonged period.** *According to the reviewer's comment, an experiment was conducted to measure the heat and current generated in the entire robotic hand. A soft ball with a diameter of 120 mm was held for 10 min while applying a fingertip force of 20 N with each of the five fingers. Fig. 7 b and Supplementary Video 2 show the measured temperature and current during the experiment. The current driving the robotic hand was almost constant (0.9 A). The measured temperature gradually increased and converged to approximately 65 °C. It operated stably even though all the fingers were holding the ball with high force. The temperature increased more rapidly than when driving with one finger, and the measured temperature was higher, but the difference was not large.*
- (4) **Repeatability test:** *Repeatability tests were performed on the robotic finger by repeating the full stretched and bent finger poses in the experimental environment displayed in Fig. 6 a. Fig. 7 c presents the force data measured during an experiment in which the force applied to the sensor for 5 s was repeated 60 times in 10 min. When the peak value of the force at each contact was defined as the contact force, the average contact force was 16.7 N, maximum error 0.4 N, and standard deviation 0.12 N in 60 contacts. Fig. 7 d presents the force data measured during an experiment in which an applied force for 1 s was repeated 300 times in 11 minutes. The average of the measured contact forces was 14.62 N, maximum error 0.66 N, and standard deviation 0.18 N. As a result, it was possible to apply a constant force without abnormal motion of the hand in numerous repeated motions. The repeatability of movement of the developed robotic finger was investigated by deriving the location of contact between the finger and sensor. During these two experiments, fingertip contact location data were derived. As shown in Fig. 7 e and f, the contact locations were derived from the measured 6-axis F/T data using the reference sensor. As a result, the standard deviation of the distance error of 60*

contact points was 0.14 mm, and the standard deviation of the error of 300 contact points was 0.099 mm. The developed robot exhibited superior repetition performance in both experiments.

- (5) **Payload test.** We conducted a payload test with dumbbells of various weights. Intuitively, it was expected that the hand would deliver superior payload performance because it is sturdy and offers a large fingertip force. Supplementary Video 2 and Fig. 7 **g** and **h** indicate that the developed hand grasps and lifts the dumbbells (7, 12, and 18 kg). It was possible to hold an 18-kg dumbbell while lifting it up and down. Because the highest payload among the investigated robotic hands was 9 kg, superior payload performance of the developed hand was demonstrated.

The corresponding content is added in Supplementary Text 5.

Q3. The integration with the sensors is very nice, even if the intermediate areas of the fingers are not covered. This limits the sensory information that can be obtained by the hand when power grasps are performed.

Most delicate tasks can be performed with fingertip sensing alone. However, as the reviewer pointed out, tactile sensing in intermediate areas is important for power grasp. We developed a small fingertip F/T sensor and integrated it compactly with the robotic hand. On the other hand, we could not obtain a tactile skin-type sensor capable of being integrated compactly to intermediate areas.

Currently, we are developing a tactile skin-type sensor that can be integrated with a robotic hand. As the reviewer is aware, the development of a skin-type sensor that is easy to use in a robot is being actively studied, and it is very difficult and important to integrate the sensor to a robot. In the future, we aim to develop artificial skin technology capable of tactile sensing and conduct research to combine it with intermediate areas such as palm of the robotic hand. We have added this content to the Discussion section.

Q4. I was impressed by the relatively low power consumption (2A at 15V), but I am not sure if it refers to a single finger or if it is the peak of hand when all the fingers exert the maximum force.

We believe that the reviewer's question about low power consumption can be

answered through the specifications of the motor used in the hand. The commercial motor (DCX 8M, Maxon motor) used has a no-load current of 2.74 mA, a maximum stall current of 0.13 A, and a nominal voltage of 12 V. Thus, when using a total of 15 motors, a total stall current of 1.95 A is required. Additionally, the driving current was verified through several experiments by measuring the current in the actual robotic hand, as mentioned in the response to Q2.

As expected, the low power consumption is secured because of the low friction of the driving part. If motor gearboxes with a high gear ratio are used, the efficiency decreases due to frictional forces as the gear stages in the gearbox increase. Since we used a low gear ratio (16:1) of the gearbox and a structure that minimizes friction by using a ball screw and a linear guide, the hand can generate a large force even at a low current. We added the motor's current information to the Methods section.

Q5. Within the paper, the authors refer to "integrated linkage-drive". What does it mean integrated linkage-drive? Is it equal to the "linkage-driven mechanism/configuration" or add something more?

The linkage-driven mechanism represents one type of mechanism constituting the developed hand. "Integrated linkage-driven," is not a mechanism and, instead, indicates that the hand is a form integrating both the actuators and the sensors. Therefore, it is expressed as "an integrated linkage-driven hand" or "an integrated linkage-driven dexterous anthropomorphic robotic hand."

Q6. "...Most linkage-driven mechanisms are composed of solid links, so it is difficult to realize the independent movement..."

Not clear here the meaning of "solid". Is it the hand presented here composed of "non solid" linkages? Is it stiff instead of "solid

We modified the sentence for greater clarity as follows.

→ *Most linkage-driven robotic fingers have realized only 1- or 2-DOF movements in which the two joints are subordinated. As a result, a linkage-driven robotic finger with three DOF has not been investigated so far, as shown in Supplementary Table 1. Additionally, to drive the PIP joint, power should be transmitted to the PIP joint*

through the MCP joint, and it is difficult to transmit the power by the movement of the MCP joint. In the case of the tendon-driven mechanism, the tendons can be easily connected to the two joints and operated independently of the joints due to its flexibility. Further details can be found in previous publications³⁷. Because links are rigid, it is difficult to solve this problem. For this reason, thus far, a 3-DOF robot finger using a linkage-driven mechanism has not been developed. Therefore, we developed a new robotic finger capable of 3-DOF movement using the proposed new linkage-driven mechanism

Q7. "...The three joints, fingertips, and palm of the robotic hand are similar to those of the human hand..." Why are you not referring to human kinematic architecture and percentile? The hand is roughly two times larger than the 50th of the male human hand in length and weight ("...maximum length of 218 mm and weight of 1.1 kg...").

According to the reviewer's comment, the kinematic architectures and percentiles of the human hand and the developed robotic hand were analyzed. Therefore, we investigated and summarized the details of the size of the human hand and the developed robot hand. In addition, we performed a workspace analysis of the proposed mechanism and compared it with the workspace of a human finger.

We added a table (Supplementary Table 2) to compare the robotic hand and human hand. The phalange lengths of an index finger, working range of each joint, palm size, and workspace volume are summarized in the table. The definition of length of a human hand is presented in Supplementary Fig. 8. A survey was conducted based on the average male hand^{48,49}. The length of the finger and the length and width of the palm of the robot were confirmed to be 5.56%, 16.79%, and 19.24% larger than those of a human, respectively. The maximum length of the robotic hand was found to be 12.93% larger than that of a human hand. The developed robotic hand was not significantly different from the human hand, as opposed to the reviewer's expectation. The explanation is added in Supplementary Text 7.

In addition, to compare the workspaces of the two hands, workspace analysis was conducted as follows. To check the workspace of the designed robotic finger, the reachable workspace of the robotic finger was analyzed as shown in Fig. 3e–g. Fig. 3e shows the 3D view of the reachable workspace composed of points that can be reached by the end point of the finger. Here, d_r is the distance between the end point of the finger and the fixed MCP joint of the finger, which is taken as the origin. The farthest points are expressed in red. In this case, both PIP and DIP of the finger at this time are stretched poses. Fig. 3f shows the

side view of the workspaces of the proposed robotic finger and human finger. As it is easy to adjust the length of the three phalanges of the robot finger, from a design perspective, the length of the phalanges of the human finger was analyzed under the same conditions as those of the robotic finger, for a close comparison. Fig. 3f shows the side view of the workspaces of the robotic finger and human finger. Here, $T_1 - T_5$, T_a , and T_{ab} indicate the trajectories of the fingers. The trajectory from T_1 to T_2 shows that the MCP joint moves from full extension (0°) to full flexion (90°) with the stretched pose of the PIP and DIP. The trajectory from T_2 to T_4 is generated when the PIP and DIP are moved while fixing the MCP joint in full flexion. T_3 represents the lowest point on the z-axis in this trajectory. T_5 is reached when the MCP joint is fully extended while maintaining full flexion of the PIP and DIP joints. The trajectory of a human finger is indicated by the dotted line. Fig. 3f shows a workspace similar to that of the two fingers.

The abduction/adduction motion is maximum (30°) when the MCP joint is in full extension and minimum (0°) when the MCP joint is in full flexion⁴⁷. For this reason, the workspace volume under the MCP joint is relatively small as shown in Fig. 3g, and a part of its lower right corner becomes sharp as shown in Fig. 3e, g.

The shape of the workspace at the bottom of the robotic finger is similar to that of the human finger. The reason is that when the MCP joint is in full flexion, the abduction/adduction movement is limited because the proximal phalanges contact with the top of the two ball screws as shown in Fig. 2e. T_a and T_{ab} are the locations of the maximum abduction/adduction movement during full extension of each joint of the human and robot fingers, respectively. The robotic hand is capable of 35° movement to perform abduction/adduction. As a result, the workspace shapes of the robotic finger and human finger are very similar, and the volume of each workspace was found to be 188740 and 119800 mm^3 , respectively.

Q8. "...it is very important to use parts with high stiffness that can sustain the force without becoming damaged..." Strength instead of stiffness? Low stiffness does not mean low strength in most of the case. Let us think for instance to a compression spring: the low stiffness is due to the long path the stress follows from the load application point to the frame (while the stress is maintained under the strength limit).

As the reviewer pointed out, stiffness is not suitable. We apologize for this oversight and have changed stiffness to strength.

Q9. Sometimes the authors have used the word configuration instead of mechanism. This is, in my opinion, confusing.

We have made changes (configuration -> mechanism) according to the reviewer's comment.

Once again, we would like to thank the reviewer for their time and effort in reviewing the previous version of our paper. This has been highly insightful and helpful to us.

Reviewers' Comments:

Reviewer #1:

Remarks to the Author:

First of all, the reviewer would like to express deep gratitude since many parts were resolved through this revision. The performances shown in the paper and videos are enough for the reviewer to be surprised, especially, fifteen active DoFs, 24N fingertip force, 1.1kg lightweight, small-sized tactile sensor, and so on. Although the scientific findings are still not many, the reviewer was so pleased to appreciate the authors' great engineering works.

Add the synergy-related parts, for example, "how to set or obtain the synergy vector when the grasping task is given"

The followings are my minor comments:

Unify the DOF and DOFs notations. The reviewer could see "22 DOF", "low DOFs", "1-DOF", "2-DOF", "one-DOF", "two-DOF" in the paper.

Unify the Equation and Eq. notation. The reviewer could see "Equation 8", "Eq. S2", and so on.

Unify the Figure and Fig. notation. The reviewer could see "Figure 4a", "Fig. 3f", and so on.

Many mathematical derivations in Kinematic Analysis Section could be moved into the supplementary material while remaining several resultant equations.

Delete the duplicated sentence. "Fig. 3f shows the side view of the workspaces of the proposed robotic finger and human finger" appears two times on the 7th page.

In the manuscript, the reviewer could discover too many "We ... " or "we ...". Please rephrase them.

The reviewer would like to recommend deleting the market-related phrases or sentences such as "In this study, we tried to maximize the mark penetration of the robotic hand in various research fields." Especially, "mark" seems to be a typo error of "market".

Please remove this sentence "In the future, the developed robotic hand is expected to achieve considerable market penetration through broadcasting media".

Reviewer #2:

Remarks to the Author:

I would like to thank the authors for their extensive response to the reviewers' comments, and for putting a great deal of effort into revising the manuscript.

Overall, I'm satisfied with their response.

The primary strength of this work is that the authors developed a dexterous, powerful and compact robot hand. I agree with the other reviewer's remarks about the scientific findings: they seem to be very narrow and limited to the Applied Mechanics community.

I will add a couple of points as a follow-up for the authors:

1- The format used for the reply letter is a bit confusing since it has been tough to distinguish between the authors' replies and the amended text.

2- Concerning the "Long-time operation of the hand" this has been conducted moving the hand in a free motion that is far unusual in a real use scenario. Durability tests are usually conducted grasping an object (eg a sponge) reaching a predefined percentage of the maximum flexion and force and for a target number of repetitions (300 K to 1 M) specifying the duty cycle.

REVIEWER COMMENTS

Reviewer: 1

Comments to the Author

First of all, the reviewer would like to express deep gratitude since many parts were resolved through this revision. The performances shown in the paper and videos are enough for the reviewer to be surprised, especially, fifteen active DoFs, 24N fingertip force, 1.1kg lightweight, small-sized tactile sensor, and so on. Although the scientific findings are still not many, the reviewer was so pleased to appreciate the authors' great engineering works.

Thank you for reviewing the manuscript and providing a concise and accurate summary of our work as well as high-quality feedback.

Q1. Add the synergy-related parts, for example, “how to set or obtain the synergy vector when the grasping task is given”

According to the reviewer's comment, we added several sentences to explain “how to set or obtain the synergy vector when the grasping task is given” as follows.

The 14 essential grasp types for grasping various objects were chosen from Feix's grasp taxonomy. The joint angle vector in every grasp type is predicted by the visual simulator of the ILDA hand. The hand synergy vector is calculated by analyzing the principal components of the change of joint angle vectors in different sizes of objects and motions from the simulator.

Human hand's actions are analyzed for specific manipulation tasks. We separated the actions into the independent components, which could be represented by one principal component of the joint angle vector. The hand synergy vectors for the manipulation task are calculated by the principal components from the motion teaching of each separated action. We teach each action of the hand and the manipulator and create a scenario for a given task. The robot environment was controlled in the task scenario.

We added the information to Control Strategy in the Methods Section.

THE FOLLOWINGS ARE MY MINOR COMMENTS:

M1. Unify the DOF and DOFs notations. The reviewer could see “22 DOF”, “low DOFs”, “1-DOF”, “2-DOF”, “one-DOF”, “two-DOF” in the paper.

We unified the notations (DOFs -> DOF, one/two/three-DOF -> 1/2/3-DOF).

M2. Unify the Equation and Eq. notation. The reviewer could see “Equation 8”, “Eq. S2”, and so on.

We unified the notations (Equation -> Eq.).

M3. Unify the Figure and Fig. notation. The reviewer could see “ Figure 4a”, “Fig. 3f”, and so on.

We unified the notations (Figure -> Fig.).

M4. Many mathematical derivations in Kinematic Analysis Section could be moved into the supplementary material while remaining several resultant equations.

According to the comment, we moved many equations into the supplementary material while remaining several resultant equations.

M5. Delete the duplicated sentence. “Fig. 3f shows the side view of the workspaces of the proposed robotic finger and human finger” appears two times on the 7th page.

We deleted the duplicated sentence.

M6. In the manuscript, the reviewer could discover too many “We ... ” or “we ...”. Please rephrase them.

We got rid of ‘we’ and rephrase them in many parts of the manuscript.

M7. The reviewer would like to recommend deleting the market-related phrases or sentences such as “In this study, we tried to maximize the mark penetration of the robotic hand in various research fields.” Especially, “mark” seems to be a typo error of “market”.

We corrected it.

M8. Please remove this sentence “In the future, the developed robotic hand is expected to achieve considerable market penetration through broadcasting media”.

We removed this sentence.

Once again, we would like to thank the reviewer for their time and effort in reviewing the previous version of our paper. This has been highly insightful and helpful to us.

Reviewer: 2

Comments to the Author

I would like to thank the authors for their extensive response to the reviewers' comments, and for putting a great deal of effort into revising the manuscript.

Overall, I'm satisfied with their response.

Thank you for reviewing the manuscript and providing a concise and accurate summary of our work as well as high-quality feedback.

The primary strength of this work is that the authors developed a dexterous, powerful and compact robot hand. I agree with the other reviewer's remarks about the scientific findings: they seem to be very narrow and limited to the Applied Mechanics community.

In the previous revision, we have clarified the purpose of our study according to the other reviewer. Consequently, we are happy that the reviewer also understand the importance of our study.

I WILL ADD A COUPLE OF POINTS AS A FOLLOW-UP FOR THE AUTHORS:

1- The format used for the reply letter is a bit confusing since it has been tough to distinguish between the authors' replies and the amended text.

In the reply letter, we will put all the explanation and the amended text. Also, we will specify the number of pages and the order of the paragraph of the revised part. Thank you for your kind help in writing the reply letter.

2- Concerning the “Long-time operation of the hand” this has been conducted moving the hand in a free motion that is far unusual in a real use scenario. Durability tests are usually conducted grasping an object (eg a sponge) reaching a predefined percentage of the maximum flexion and force and for a target number of repetitions (300 K to 1 M) specifying the duty cycle.

We are considered the comment on additional reliability test of the developed hardware. The experiment of repeatedly holding and placing an object such as a sponge suggested by the reviewer is considered a good example to show the reliability of the

developed robotic hand. We performed the analysis by performing five additional experiments [(1)Long-time operation..., (2)Constant fingertip force..., (3)Grasping a ball..., (4)Repeatability, (5)Payload test] for reliability testing in the previous revision.

(1)Long-time operation... is simply repeating several motions many times. However, during the experiment, the motors integrated into the robotic hand were running at full speed with the relatively high power consumption. The experiment on repetitive motion and force suggested by reviewers were performed through the (4) repeatability test. We believe that the performance evaluation at the research level has been sufficiently performed. However, it is not enough for the target number of repetitions (300k~1M) suggested by the reviewer. In the future, we plan to conduct experiments that satisfy the target number to prepare for actual commercialization. Thanks for the reviewer's advice in terms of practical use of the robotic hand.

Once again, we would like to thank the reviewer for their time and effort in reviewing the previous version of our paper. This has been highly insightful and helpful to us.